# Counterbalancing Learning and Strategic Incentives in Allocation Markets

**Jamie Kang**
Stanford University
kangjh@stanford.edu

**Faidra Monachou**
Stanford University
monachou@stanford.edu

**Moran Koren**
Harvard University
me@mkoren.org

**Itai Ashlagi**
Stanford University
iashlagi@stanford.edu

## Abstract

Motivated by the high discard rate of donated organs in the United States, we study an allocation problem in the presence of learning and strategic incentives. We consider a setting where a benevolent social planner decides whether and how to allocate a single indivisible object to a queue of strategic agents. The object has a common true quality, good or bad, which is ex-ante unknown to everyone. Each agent holds an informative, yet noisy, private signal about the quality. To make a correct allocation decision the planner attempts to learn the object quality by truthfully eliciting agents' signals. Under the commonly applied sequential offering mechanism, we show that learning is hampered by the presence of strategic incentives as herding may emerge. This can result in incorrect allocation and welfare loss. To overcome these issues, we propose a novel class of incentive-compatible mechanisms. Our mechanism involves a batch-by-batch, dynamic voting process using a majority rule. We prove that the proposed voting mechanisms improve the probability of correct allocation whenever agents are sufficiently well informed. Particularly, we show that such an improvement can be achieved via a simple greedy algorithm. We quantify the improvement using simulations.

## 1   Introduction

This paper contributes to the growing literature in the intersection of learning and mechanism design, by considering a fundamental allocation problem in the presence of learning and strategic agents. In our setting, a planner needs to allocate a single indivisible object to at most one among several agents waiting in a queue. The object's true quality is either high ("good") or low ("bad"), but is ex-ante unknown to both the planner and the agents. Agents are *privately informed*, in the sense that each receives a private signal that is informative about the true quality of the object, and *strategic* since they choose to accept or reject the object based on the action maximizing their expected utility. In this framework, agents thus engage in a *social learning* process as they observe the actions of their predecessors but not their private signals. The planner's goal is to design an incentive-compatible mechanism that optimizes *correctness*, i.e., maximizes the ex-ante accuracy of allocation.

This problem finds a natural application in the allocation of donated organs to patients on a national waitlist. The median waiting time for a first kidney transplant in the U.S. is around 3.6 years [25], whereas the rate at which deceased donor kidneys are offered to patients but eventually discarded has been growing steadily [23], exceeding 21.2% in 2020 [30]. The reasons leading to high discard rates – even of seemingly high-quality organs – remain unclear. Clinical studies attribute this increase

35th Conference on Neural Information Processing Systems (NeurIPS 2021).

to herding behavior and inefficiency in allocation mechanisms [11, 29]. Improving the allocation outcome of these markets is undoubtedly crucial as it could save thousands of future lives.

Towards understanding the source of high discard rates, our framework highlights the often overlooked role of social learning. In most cases, when offered an organ, patients not only have access to publicly available data about the organ (e.g., donor's age, organ size) but also consult with their own physicians for private opinions. Each physician may assess the same organ differently based on their prior experience and/or medical expertise. Empirical works [13, 31] have recently proposed an explanation that patients rationally ignore their own private information (e.g., a physician's medical assessment) about the quality of an organ and follow the preceding patients' rejections.

Our proposed model captures this phenomenon and serves as a theoretical sandbox for the design and assessment of alternative allocation policies. We theoretically show that sequentially offering the object to agents – which is a policy commonly used in practice – can result in poor correctness, since the rejections by the few initial agents can cause a cascade of rejections by the subsequent agents.

To address this issue, we introduce a novel class of incentive-compatible allocation mechanisms characterized by a *batch-by-batch, dynamic voting* process under a simple majority rule. Our proposed greedy mechanism dynamically partitions the list into batches of varying sizes. Then, it sequentially offers the object to each batch until the object is allocated to some agent. The planner privately asks each agent in the current batch to either opt in or opt out. If the majority of the current batch vote to opt in, then the planner randomly allocates the object to one of the current agents who opted in. If the mechanism exhausts the list without success, the object is discarded. Interestingly, several Organ Procurement Organizations (OPOs) already offer organs in batches, although no voting is used;[1] the reasons are both to speed up the allocation process and to avoid herding.

In our main result, we show that if private signals are at least as informative as the prior, there always exists an incentive-compatible voting mechanism that strictly improves correctness and thus helps avoid unwanted discards. Furthermore, we suggest a simple, greedy algorithm to implement such a mechanism. For all other values of private signal precision, we establish that there is no incentive-compatible voting mechanism. Moreover, every voting mechanism results in the same correctness as the sequential offering mechanism.

Incentive-compatible voting mechanisms have several interesting properties. First, the batch size increases after each failed allocation attempt. The intuition is simple. Failed allocations make the current belief about the object quality more pessimistic. Therefore, the planner increases the batch size to ensure that the object is allocated only if a sufficient number of agents receive positive signals while each agent's decision remains pivotal. Second, offering to another batch after a failed attempt strictly improves correctness (compared to sequential offering). As our simulations showcase, this improvement is significant and generally becomes larger as the prior and signal precision grow; in fact, a voting mechanism with just two batches significantly outperforms sequential offering. Finally, we find that the prior and signal precision have opposite effects on correctness: the correctness tends to decrease as the prior increases but the marginal effect of signal precision is positive. An intuitive explanation is as follows. Since the optimal batch size decreases as the prior increases, the probability to incorrectly allocate a bad object increases. However, as the signal precision improves, the magnitude of the error decreases since agents' signals become more reliable.

We highlight the *tension between agents' strategic incentives and the planner's learning goal*. From the planner's perspective, agents' private signals contain valuable information about the unknown quality of the object. By increasing the number of collected data points, the planner can improve her confidence about the true quality of the object. Agents' strategic incentives, however, impose a constraint on the maximum size of the batch that allows the truthful elicitation of agents' signals: as the batch size increases, the chance to be pivotal decreases which in turn reduces the incentive for an agent with a negative signal to be truthful. Our proposed voting mechanisms enable the planner to crowdsource agents' private information in a simple, truthful, and effective manner.

**Related literature.** Our model draws upon seminal social learning papers [7, 8, 28] and their extensions in various game structures [1, 2, 14, 24]. Several empirical studies also examine social learning in organ allocation settings [13, 31]. However, none of these works consider the existence of a planner that controls the flow of information. Furthermore, our proposed mechanism is inspired by the voting literature (e.g., [6, 12]); our notion of correctness is adopted from [3]. Finally, our paper

---

[1]Based on authors' personal communication. See, e.g., [21] for allocation in batches.

is loosely connected to information design (e.g., [10, 18, 26, 27]) and Bayesian exploration (e.g., [15–17, 20, 22]). In contrast to our model, this literature does generally not consider an underlying social learning process. We include an extended overview of the related literature in Appendix A.

## 2 Model

We consider a model in which a social planner wishes to allocate a single indivisible object of unknown quality to privately informed agents in a queue.

**Object and agents.** The object is characterized by a fixed *quality* $\omega \in \{G, B\}$, where $G$ and $B$ denote a *good* and *bad* quality, respectively. The true quality $\omega$ of the object is ex-ante unknown to both the planner and agents; however, both share a common prior belief $\mu = \mathbb{P}(\omega = G) \in (0, 1)$.

Agents are waiting in a queue; we denote by $i$ the agent in position $i$. Each each agent knows his own position. For simplicity, we assume that the queue consists of an arbitrarily large number $I$ of agents. Each agent $i$ has a private binary *signal* $s_i \in \mathcal{S} \triangleq \{g, b\}$ that is informative about the true quality of the object $\omega$. Each $s_i$ is identically and independently distributed conditional on $\omega$. Furthermore, each $s_i$ is aligned with $\omega$ with probability $q = \mathbb{P}(s_i = g \mid \omega = G) = \mathbb{P}(s_i = b \mid \omega = B)$, where the signal precision $q \in (0.5, 1)$ is commonly known.[2]

Agents are risk-neutral. The utility of the agent who receives the object is 1 if $\omega = G$ and $-1$ if $\omega = B$ (This symmetry simplifies the exposition and analysis but does not qualitatively change the results.). Any agent who does not receive the object, because he either declines or is never offered, receives a utility of 0. As a tie-breaking rule we assume that indifferent agents always decline.

**Voting mechanisms.** The planner designs a mechanism which potentially asks (a batch of) agents to report their private signals, and based on their report, decides whether and how to allocate the object. We consider a class of voting mechanisms denoted by $\mathcal{V}$. A voting mechanism $V \in \mathcal{V}$ offers the object sequentially to odd-numbered batches of agents, where each batch size may be set dynamically based on the information from previous batches. In any given batch, if the majority of agents vote to opt in, the object is allocated uniformly at random to one of the current agents who opted in.

Formally, a *voting mechanism* $V_{\{\pi_j\}_{j=1}^{\infty}}$ is defined by a sequence of functions $\pi_1, \pi_2, ...,$ such that for each batch $j$, the corresponding function $\pi_j : [0, 1] \to \mathbb{N}$ maps a current belief $\mu_{j-1}$ about the object quality $\omega$ to a batch size $K_j$ (we let $\mu_0 = \mu$). The mechanism begins with offering to the first batch of agents, who are in positions $1, \ldots, \pi_1(\mu)$. If the object is not allocated to batch $j - 1$, it is offered to the agents in the next $\pi_j(\mu_{j-1})$ positions.

When batch $j$ is offered the object, each agent $i$ in batch $j$ chooses an action (a vote) $\alpha_i \in \mathcal{A} \triangleq \{y, n\}$, where $y$ and $n$ correspond to opting in ("yes") and opting out ("no"), respectively. If $Y_j$, the number of opt in votes from batch $j$, constitutes the *majority* of batch $j$, i.e., $Y_j \geq \lceil \frac{\pi_j(\mu_{j-1})}{2} \rceil$, the object is allocated uniformly at random among agents in batch $j$ who opt in. To simplify exposition, we sometimes denote by $K_j$ the ex-post size of batch $j$ (while omitting the dependency on the current belief $\mu_{j-1}$). We assume that the agents in batch $j$ as well as the planner observe all the votes of agents in the previous batches $j' < j$, and this is commonly known. After any batch $j$ and given a current prior $\mu_{j-1}$, the belief $\mu_j$ shared by the planner and agents is updated recursively in terms of the realized batch size $K_j$ and voting outcome $Y_j$ as

$$\mu_j \triangleq \mathbb{P}(\omega \mid \mu_{j-1}, K_j, Y_j) = \frac{\mu_{j-1} q^{Y_j}(1-q)^{K_j - Y_j}}{\mu_{j-1} q^{Y_j}(1-q)^{K_j - Y_j} + (1 - \mu_{j-1}) q^{K_j - Y_j}(1-q)^{Y_j}}. \quad (1)$$

A commonly applied voting mechanism is a *sequential offering* mechanism, which offers the object to one agent at a time. Another type of voting mechanism is *single-batch voting* mechanisms, which offer the object only once; thus, the object is discarded if the majority in the batch choose to opt out.

Let $u_i(\alpha_i; s_i)$ denote the expected utility of an agent who receives signal $s_i \in \mathcal{S}$ and takes action $\alpha_i \in \mathcal{A}$. A voting mechanism $V \in \mathcal{V}$ is *incentive-compatible* (IC) if $u_i(y; g) \geq u_i(n; g)$ and $u_i(n; b) \geq u_i(y; b)$. Thus, agent $i$ gets higher utility from opting in than opting out when he has received signal $s_i = g$, but prefers to opt out when $s_i = b$.

---

[2]The condition $q \in (0.5, 1)$ is without loss of generality. The common prior belief $\mu$ and signal precision $q$ are standard in the social learning literature [7, 8, 28].

**Correctness.** The planner's goal is to maximize the probability of a correct allocation outcome, or *correctness*; the allocation outcome is correct if it allocates a good object or does not allocate a bad object. Formally, under a voting mechanism $V \in \mathcal{V}$, the planner's decision whether to allocate the object or not is denoted by the random variable $Z = \{0, 1\}$.

A mechanism $V \in \mathcal{V}$ achieves correctness $c(V)$ if

$$c(V) \triangleq \mathbb{E}_V \left[ \mathbb{I}(\omega = G \cap Z = 1) + \mathbb{I}(\omega = B \cap Z = 0) \right].$$

Since we allocate a single object, a cascade can only form on the opt out action. Therefore, maximizing correctness is equivalent to maximizing social welfare.

**Upper bound on correctness.** A natural benchmark on correctness would be the optimal solution in a setting where agents are not strategic and thus are willing to truthfully reveal their private signal to the planner. In the absence of strategic incentives, the planner's optimal solution would be to (i) ask *all* agents in the queue to reveal their private signals, (ii) compute her posterior belief based on the gathered information, and (iii) allocate the object to a random agent if and only if her posterior exceeds 0.5. Equivalently, as we show in Lemma A.1 in Appendix A, the planner allocates the object if and only if the number of positive signals is at least $\underline{y} \triangleq 0.5 \log\left(\frac{1-\mu}{\mu}\right) \left(\log\left(\frac{q}{1-q}\right)\right)^{-1} + 0.5\,I$, where $I$ is the number of agents in the queue. In that case, we can show that correctness equals

$$\mathbb{P}\left(X_I \geq \underline{y}\right), \tag{2}$$

where $X_I$ is a Binomial$(I, q)$ random variable (see also Lemma 3). As $I$ grows, correctness approaches 1. Importantly, this upper bound is not achievable by an incentive-compatible mechanism.

**Discussion of modeling assumptions.** The model is based on several modeling choices that we discuss next. First, the assumption that voting outcomes are fully revealed to subsequent batches is made for simplicity. Dropping this knowledge complicates the analysis substantially: one should compute a posterior belief of an agent by taking expectations of possible outcomes that led to them being offered an object. Second, assuming that batches are always odd-numbered is of technical nature. To attain incentive compatibility, our analysis relies on the monotonic behavior of the upper and lower bounds on the prior. This result depends on a probabilistic argument about binomial distributions (Lemma D.2), which fails to hold if batches can be both of odd and even sizes.[3] Finally, we consider the allocation of just a single object rather than a setting in which objects arrive and are allocated sequentially. In particular we abstract away from dynamic incentives and restrict attention to the learning aspect. We leave the interaction between the two for future work.

## 3 Herding in the Sequential Offering Mechanism

In this section we analyze the commonly applied sequential offering mechanism, which will serve as a benchmark for our results. Recall that the *sequential offering* mechanism, namely $V_{\text{SEQ}}$, offers the object to each agent one-by-one; the first agent that is offered the object and opts in receives it.

As we establish below, the sequential offering mechanism has two important drawbacks: it is not incentive-compatible and achieves poor correctness. This is due to a cascade of actions (i.e., herding) that takes place after a few decisions. Specifically, a cascade of opt out actions begins after a couple of initial agents opt out, which ultimately leads to the discard of the object. The number of initial opt out actions needed to instigate this cascade-led discard depends on the prior $\mu$. The following result formalizes this. A detailed analysis of the benchmark mechanism can be found in Appendix B together with the omitted proofs of the section.

**Lemma 1.** *Under* $V_{\text{SEQ}}$, *the object is allocated if and only if: (i)* $\mu > q$, *(ii)* $\mu \in [1/2, q]$ *and either* $s_1 = g$ *or* $s_2 = g$, *(iii)* $\mu \in [1-q, 1/2]$ *and* $s_1 = g$. *Otherwise, the object is discarded.*

A direct implication of this result is that the outcome of the mechanism is determined in the first two offers. Regardless of the values of $\mu$, $q$ and the object's true quality, the object is always discarded if it is rejected by the first two agents. Thus the result below follows.

**Proposition 1.** *Under the sequential offering mechanism,* $V_{\text{SEQ}}$:
   *(i) At most the two initial agents in the queue determine the allocation outcome: if the first two agents decline, then the object will be always discarded.*

---

[3]This issue can be addressed by introducing a random tie-breaking rule with appropriate weights.

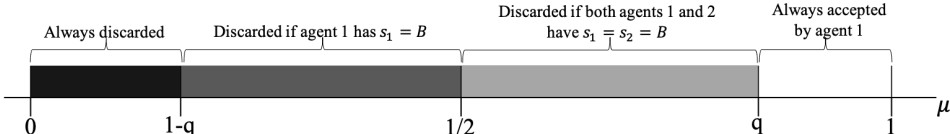

Figure 1: Allocation outcome of the sequential offer mechanism ($V_{\text{SEQ}}$) based on the value of prior $\mu \in (0, 1)$ with respect to signal precision $q$. For low $\mu$, the object is rejected by all agents, leading to discard. For intermediate $\mu$, the allocation outcome depends on the initial two agents' private signals.

(ii) *The correctness equals* $c(V_{\text{SEQ}}) = \begin{cases} \mu & \textit{for } \mu > q \\ 2\mu q(1-q) + q^2 & \textit{for } \mu \in [1/2, q] \\ q & \textit{for } \mu \in [1-q, 1/2) \\ 1-\mu & \textit{for } \mu < 1-q. \end{cases}$

The result qualitatively suggests that the first few[4] agents in the queue have the power to decide whether the object will be allocated. This implies that these few agents can inadvertently undermine the welfare of the rest of the agents. Most importantly, the planner can never learn from the private information of the remaining $I - 2$ agents to make better allocation decisions. In the context of organ transplants, this herding behavior prevents the planner from allocating good-quality organs, leading to a high discard rate. This may in turn lead to patients' longer waiting times and health deterioration.

## 4  Voting Mechanisms

Recall from Equation (2) that in the absence of strategic incentives, the planner asks as many agents as possible and makes her allocation decision based on these (truthful) private signals. On the other hand, as we have seen with $V_{\text{SEQ}}$, when agents are strategic, the presence of their incentives introduces a constraint for the planner, which in turn sets an upper bound on the number of agents the planner can truthfully learn from. Under the benchmark mechanism $V_{\text{SEQ}}$, for example, the planner can elicit at most two truthful signals, which may prevent her from taking a correct allocation decision.

Motivated by these shortcomings, we propose a new class of voting mechanisms and examine the incentive-compatible mechanisms within this class. We show that there always exists an incentive-compatible voting mechanism that improves correctness as long as private signals are more informative than the common prior belief; otherwise, none of the voting mechanisms are incentive-compatible and all achieve correctness equal to that of $V_{\text{SEQ}}$. Theorem 1 is our main result.

**Theorem 1.** *For any $\mu < q$, there exists a voting mechanism $V \in \mathcal{V}$ that is incentive-compatible and improves correctness compared to the sequential offering mechanism $V_{\text{SEQ}}$. For $\mu \geq q$, there is no incentive-compatible voting mechanism and any $V \in \mathcal{V}$ achieves the same correctness as $V_{\text{SEQ}}$.*

We prove Theorem 1 in several steps. In Section 4.1, we begin by defining the simplest voting mechanisms, the *single-batch voting mechanisms,* where the mechanism offers the object to one batch only and terminates afterwards. Specifically, we characterize the optimal mechanisms among all incentive-compatible single-batch voting mechanisms. Then in Section 4.2, we utilize these results to develop a simple greedy algorithm to construct an incentive-compatible, correctness-improving voting mechanism with potentially *multiple* batches.

### 4.1  Warm up: Optimal Single-batch Voting Mechanisms

As a warm up, we begin with the study of one of the simplest voting mechanisms: the single-batch voting mechanisms. With a slight abuse of notation, we use the following definition.

**Definition 1.** Let $V_{\{K\}} \in \mathcal{V}$ be the *single-batch voting mechanism* where the object is offered to only one batch of the initial $K$ agents in the queue.

Unlike $V_{\text{SEQ}}$, the single-batch voting mechanism $V_{\{K\}}$, as well as any other voting mechanism, ensures that the object is allocated if and only if the majority of agents (in the batch) opt in. As our

---

[4]We have assumed that agents share a constant prior $\mu$. In practice, each agent's prior might be drawn i.i.d. from some common distribution with support $(0, 1)$. Our qualitative insights still extend to this case: a larger but finite number of initial agents determines the allocation outcome and herding still occurs.

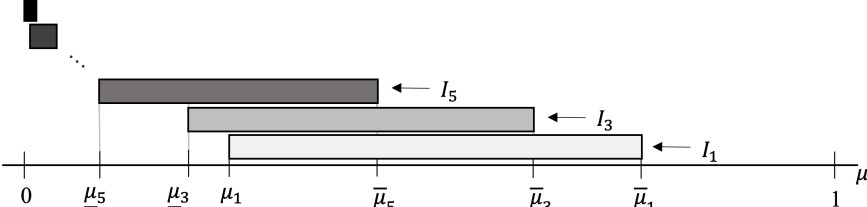

Figure 2: Interval of incentive-compatible prior $\mathcal{I}_K$ for different batch sizes $K \in \{1, 3, \ldots\}$ as described in Lemma 2. The intervals $\mathcal{I}_K$ are decreasing (Lemma C.1) with $K$ and the consecutive intervals $\mathcal{I}_K$ and $\mathcal{I}_{K+2}$ overlap (Lemma C.2) with each other.

results in this section illustrate, the extent to which this policy incentivizes agents to vote truthfully depends on the values of the prior $\mu$ and batch size $K$.

In Lemma 2, we identify the necessary and sufficient conditions for $V_{\{K\}}$ to be incentive-compatible. In Lemma 3, we show its existence conditional on the informativeness of signals and characterize the correctness it achieves. In Proposition 2, we establish that among all incentive-compatible $V_{\{K\}}$, correctness is maximized when the incentive compatibility constraint is binding at $K = \overline{K}(\mu)$. Finally in Lemma 4, we provide simple comparative statics of this optimal batch size. In the next section, we will extend these results to voting mechanisms possibly with multiple batches.

First, for any batch size $K$, let

$$\mathcal{I}_K \triangleq \left( 1 - \mathbb{P}\left( X_K \geq \frac{K+1}{2} \right), \frac{q^2 \left( 1 - \mathbb{P}\left( X_K \geq \frac{K+1}{2} \right) \right)}{q^2 \left( 1 - \mathbb{P}\left( X_K \geq \frac{K+1}{2} \right) \right) + (1-q)^2 \mathbb{P}\left( X_K \geq \frac{K+1}{2} \right)} \right)$$

be some interval of prior where $X_K \sim \text{Binomial}(K, q)$. Moreover, let $\overline{K}(\mu)$ (respectively, $\underline{K}(\mu)$) be the maximum (respectively, minimum) batch size $K$ such that $\mu \in \mathcal{I}_K$. Then the incentive compatibility of $V_{\{K\}}$ can be characterized in terms of the interval $\mathcal{I}_K$, or equivalently, the batch size bounds $\overline{K}(\mu)$ and $\underline{K}(\mu)$:

**Lemma 2.** $V_{\{K\}}$ *is incentive-compatible if and only if (i)* $\mu \in \mathcal{I}_K$, *or equivalently (ii)* $\mu < q$ *and* $\underline{K}(\mu) \leq K \leq \overline{K}(\mu)$.

The proof of the lemma can be found in Appendix C together with the rest of the proofs in this section. A sketch of the proof is as follows. To show condition (i), for a fixed $K$, let $\mathcal{G}_K$ (respectively, $\mathcal{B}_K$) be the probability that the object gets allocated to some agent in the batch conditional on the true quality being good (respectively, bad). We compute that

$$\mathcal{G}_K = \sum_{y = \frac{K+1}{2}}^{K} \frac{1}{y} \binom{K-1}{y-1} q^{y-1}(1-q)^{K-y} \qquad \mathcal{B}_K = \sum_{y = \frac{K+1}{2}}^{K} \frac{1}{y} \binom{K-1}{y-1} q^{K-y}(1-q)^{y-1}.$$

Next, we use $\mathcal{G}_K$ and $\mathcal{B}_K$ to rewrite the incentive compatibility constraints as $\mu q \mathcal{G}_K - (1-\mu)(1-q)\mathcal{B}_K \geq 0$ and $\mu(1-q)\mathcal{G}_K - (1-\mu)q\mathcal{B}_K \leq 0$. Given that the left-hand sides of both inequalities monotonically increase with $\mu$, there exist some thresholds of prior, namely $\underline{\mu}_K \in (0,1)$ and $\overline{\mu}_K \in (0,1)$, that solve the indifference conditions: $\frac{\underline{\mu}_K}{1-\underline{\mu}_K} = \frac{1-q}{q} \frac{\mathcal{B}_K}{\mathcal{G}_K}$ and $\frac{\overline{\mu}_K}{1-\overline{\mu}_K} = \frac{q}{1-q} \frac{\mathcal{B}_K}{\mathcal{G}_K}$. Based on this transformed system of equations, we are able to characterize the feasible region $(\underline{\mu}_K, \overline{\mu}_K)$ for the prior $\mu$ which turns out to be precisely $\mathcal{I}_K$. To show the equivalent condition (ii), we utilize some useful properties of the interval $\mathcal{I}_K$: first, its endpoints are weakly decreasing in $K$ (Lemma C.1), and second, it is overlapping (i.e., $\mathcal{I}_K \cap \mathcal{I}_{K+2} \neq \phi$) (Lemma C.2).

Condition (ii) in terms of batch size $K$ provides a straightforward intuition. Larger batch sizes make agents more confident that, if the object is allocated, it is likely that the quality is good. For the same reason, however, a batch size that is *too large* incentivizes everyone to opt in, including those with negative signals. Borrowing the terminology from the voting literature, $\overline{K}(\mu)$ is designed to make each agent's decision *pivotal*: $\overline{K}(\mu)$ is the maximum batch size that guarantees incentive compatibility. On the other hand, if the batch size is *too small*, then the allocation decision depends on learning from a sample size that is too small to offer reliable information, making agents with

positive signals reluctant to opt in. Thus, $\underline{K}(\mu)$ is the minimum batch size that ensures incentive compatibility. Furthermore, Lemma 2 gives rise to the following observation.

**Lemma 3.** *Suppose $\mu < q$. Then, an incentive-compatible $V_{\{K\}}$ always exists and achieves correctness $c\left(V_{\{K\}}\right) = \mathbb{P}\left(X_K \geq \frac{K+1}{2}\right)$, where $X_K \sim Binomial(K, q)$.*

Lemma 3 shows that if each signal is at least as informative as the prior $\mu$, there is a single-batch voting mechanism that elicits information in a truthful manner. Another implication of the lemma is that the correctness of an incentive-compatible $V_{\{K\}}$ can be expressed as the tail distribution of a Binomial$(K, q)$ random variable. This tail distribution can be interpreted as the probability that the majority of votes are aligned with the true quality. Given the definition of correctness, this property is thus particularly intuitive.

An additional observation is that the tail distribution $\mathbb{P}\left(X_K \geq \frac{K+1}{2}\right)$ increases with $K$ (see Lemma D.2 in Appendix D). Consequently, given $\mu$, the upper size limit $\overline{K}(\mu)$ maximizes correctness among all possible values of $K$ that conserve the incentive compatibility of $V_{\{K\}}$. We formalize this property below.

**Proposition 2.** *Suppose $\mu < q$. Then the batch size $K = \overline{K}(\mu)$ maximizes correctness among all incentive-compatible $V_{\{K\}}$.*

Hence the planner will choose the largest batch size that the agents' incentives allow. As we establish below, the optimal batch size behaves monotonically with respect to the prior $\mu$.

**Lemma 4.** $\overline{K}(\mu)$ *weakly decreases with $\mu$.*

The logic here is simple. If the sentiment around the object quality is optimistic (i.e., $\mu$ is high), then it is harder to keep the agents with negative signals truthful, requiring a tighter upper bound on the batch size (lower $\overline{K}(\mu)$). We discuss additional comparative statics through simulations in Section 5.

Finally, note that the results above focused on regimes where $\mu < q$. This a natural setting which guarantees that the signal is at least as informative as the prior belief. For the complementary case where $\mu \geq q$, we show that there is no incentive-compatible voting mechanism (see Lemma C.3 in Appendix C). Intuitively, the lack of informativeness of the signal induces the agents to lose trust in their private signals, and as a result creates no incentives to behave truthfully. In that case, under any $V_{\{K\}}$, all agents will opt in and thus the object will always be allocated. Notice that this outcome is equivalent to the outcome under $V_{\text{SEQ}}$ in terms of correctness.[5]

## 4.2 Improving Correctness via Greedy Voting Mechanisms

Next we utilize the results of Section 4.1 to characterize how to dynamically choose batch sizes of a voting mechanism (potentially with multiple batches) to improve correctness, while balancing the agents' incentives. Further, we implement this mechanism via a simple greedy algorithm.

The first key technical observation is that, from the perspective of the agents in batch $j$, the voting mechanism $V_{\{\pi_j\}_{j=1}^{\infty}}$ is equivalent to a single-batch voting mechanism $V_{\{K_j\}}$ with realized common belief $\mu_{j-1}$ and batch size $K_j = \pi_j(\mu_{j-1})$. Naturally, since both the planner and agents observe the voting results from previous batches, no informational asymmetry arises between them.

**Lemma 5.** *A voting mechanism $V_{\{\pi_j\}_{j=1}^{\infty}}$ is incentive-compatible if and only if for any batch $j$, current belief $\mu_{j-1}$ and batch size $K_j \triangleq \pi_j(\mu_{j-1})$, $V_{\{K_j\}}$ is incentive-compatible.*

Hence, to ensure the incentive compatibility of $V_{\{\pi_j\}_{j=1}^{\infty}}$, it is sufficient to choose each batch size $K_j$ myopically by solving a single-batch voting mechanism design problem (with updated belief $\mu_{j-1}$ instead of $\mu$). This suggests the following greedy scheme.

**Definition 2.** For any $J \in \mathbb{N}$, let $V_{\text{GREEDY}}^J \in \mathcal{V}$ be the voting mechanism that offers the object to $J$ batches unless either it is allocated or the list is exhausted. Formally, for each batch $j$ and realized belief $\mu_{j-1}$, the batch sizes are chosen as $K_j = \pi_j(\mu_{j-1}) = \begin{cases} \overline{K}(\mu_{j-1}) & \text{for } j \leq J \\ 0 & \text{for } j > J. \end{cases}$

---

[5]The only difference is that under $V_{\{K\}}$, the object is always allocated to an agent in the batch uniformly at random, whereas under $V_{\text{SEQ}}$ it is always allocated to agent 1.

Building upon Lemma 5 and results from Section 4.1, we establish three key properties of $V_{\text{GREEDY}}^J$.

**Proposition 3.** *For any $\mu < q$ and $J$, $V_{\text{GREEDY}}^J$ has the following properties:*

   *(i) $V_{\text{GREEDY}}^J$ is incentive-compatible;*
   *(ii) Ex-post batch sizes satisfy $K_{j'} \leq K_j$ for any $j' < j \in [J]$;*
   *(iii) $c\left(V_{\text{GREEDY}}^J\right)$ strictly increases with $J$.*

To understand part (ii), suppose that the object had been offered to batch $j$ but was not allocated. Then, the current belief naturally decreases, that is $\mu_j < \mu_{j-1}$, as the majority of batch $j$ has voted to opt out. By Lemma 4, a more pessimistic belief about the object quality results in a larger optimal batch size $\overline{K}(\cdot)$. Hence, it follows that $\overline{K}(\mu_{j-1}) \leq \overline{K}(\mu_j)$. Part (iii) of the proposition suggests that adding an additional batch improves correctness. The intuition is that an additional batch allows the planner to collect and learn from more information about the object. Consequently, the planner would like to keep on offering until either the object is allocated or there are not enough agents left in the queue. We formally define this mechanism below and describe a simple algorithm to implement it.

**Definition 3.** Let $V_{\text{GREEDY}} \in \mathcal{V}$ be the *greedy voting mechanism* such that for each batch $j$ and the realized belief $\mu_{j-1}$, the batch sizes are chosen as $K_j = \pi_j(\mu_{j-1}) = \overline{K}(\mu_{j-1})$.

---

**Algorithm 1:** Implementation of $V_{\text{GREEDY}}$

---

Initialize batch $j = 1$, belief $\mu_0 = \mu$, batch size $K_1 = \overline{K}(\mu_0)$;
**while** $I \geq K_1 + \ldots + K_j$ **do**
    Collect votes from the top $K_j$ remaining agents;
    **if** $Y_j \geq \lceil \frac{K_j}{2} \rceil$ **then**
        Allocate the object uniformly at random among agents in batch $j$ who opted in;
    **else**
        Update belief $\mu_j$ using Equation (1) and next batch size $K_{j+1} = \overline{K}(\mu_j)$;
        Set $j \leftarrow j + 1$;

---

Note that a major advantage of $V_{\text{GREEDY}}$ is its design simplicity: the maximum number of batches, $J$, does not have to be predefined.

Finally, we are ready to prove Theorem 1. The main idea is that using the correctness $c(V_{\text{SEQ}})$ of the sequential offering mechanism from Proposition 1, we can show that for every $\mu < q$, the planner can achieve $c\left(V_{\text{GREEDY}}^J\right)$ that exceeds $c(V_{\text{SEQ}})$, by appropriately setting $J$ (full proof in Appendix C.3). Furthermore, using Proposition 3 (part (iii)), we establish that the improvement described in Theorem 1 can be also achieved by $V_{\text{GREEDY}}$.

**Corollary 1.** *For any $\mu < q$, $V_{\text{GREEDY}}$ is an incentive-compatible (multi-batch) voting mechanism that improves correctness in comparison to the sequential offering mechanism $V_{\text{SEQ}}$.*

## 5 Simulations

In this section we use simulations to complement our theoretical results.

**Optimal batch size.** In Figure 3, we study the optimal batch size $\overline{K}(\mu)$ (Proposition 2) of the single-batch voting mechanism $V_{\{K\}}$; by construction, the optimal $V_{\{K\}}$ is equivalent to $V_{\text{GREEDY}}^1$. In our numerical analysis, we examine $\overline{K}(\mu)$ for all priors $\mu \in (0, 1)$ and signal precision $q \in \{0.6, 0.7, 0.8\}$.

We make several observations. First, as guaranteed by Lemma 4, the optimal batch size $\overline{K}(\mu)$ decreases as the prior $\mu$ increases. As we discussed in greater detail in Section 4.1, the incentive compatibility constraint binds at lower values of $\overline{K}(\mu)$. For very high values of $\mu$, that is $\mu \geq q$, Theorem 1 establishes that it is impossible to achieve incentive compatibility for any batch size, which explains the discontinuity at $\mu = q$ in each curve.

Second, for lower values of $\mu$ (approximately for values $\mu < 0.5$), the marginal effect of $q$ on $\overline{K}(\mu)$ is negative but diminishes as $\mu$ grows. Indeed, as Figure 3 illustrates, the signal precision $q$ plays an important role for lower values of $\mu$. In particular, if $\mu$ is low and $q$ is not significantly informative (e.g., $q = 0.6$), the planner has a priori low confidence in the object's quality. Similarly, agents have strong incentives to opt out regardless of their signal. Thus, the planner needs a larger batch size.

**Correctness evaluation and comparison.** In Figure 4, we evaluate the correctness $c(V)$ of four different mechanisms $V \in \{V_{\text{GREEDY}}^1, V_{\text{GREEDY}}^2, V_{\text{SEQ}}, V_{\text{ALL}}\}$ for $q \in \{0.6, 0.7, 0.8\}$ and $\mu \in (0, 1)$. For

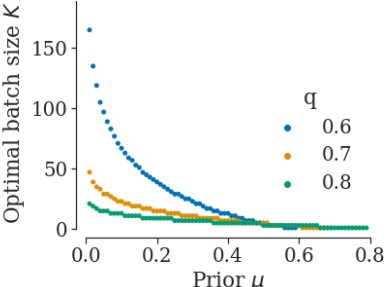

Figure 3: Via simulations, we compute the optimal incentive-compatible batch size $\overline{K}(\mu)$ for all possible priors $\mu \in (0, 1)$ for three regimes: $q \in \{0.6, 0.7, 0.8\}$. In all regimes, $\overline{K}(\mu)$ decreases as $\mu$ increases. For low values of $\mu$, higher $q$ implies lower $\overline{K}(\mu)$.

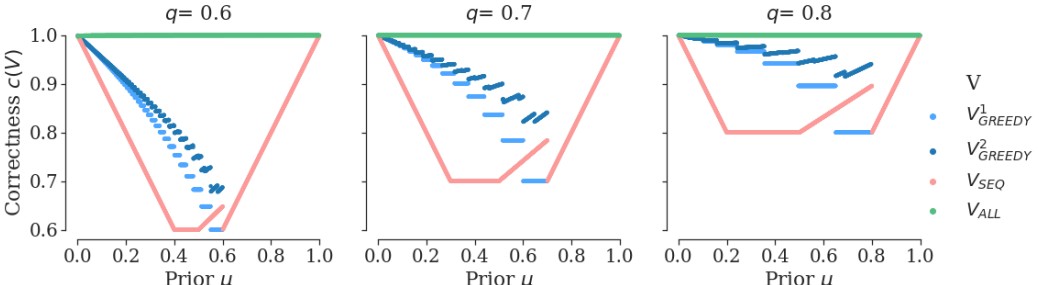

Figure 4: We compare the correctness $c(V)$ of mechanisms $V \in \{V^1_{\text{GREEDY}}, V^2_{\text{GREEDY}}, V_{\text{SEQ}}, V_{\text{ALL}}\}$ in a setting with population size $I = 345$. $V_{\text{ALL}}$ represents the optimal mechanism in the absence of strategic incentives, which achieves near perfect correctness close to 1 (see Equation (2)). The remaining $V^1_{\text{GREEDY}}, V^2_{\text{GREEDY}}$, and $V_{\text{SEQ}}$ assume strategic agents. We compute $c(V)$ for all $\mu \in (0, 1)$ and three different regimes $q \in \{0.6, 0.7, 0.8\}$. In all regimes, $V^1_{\text{GREEDY}}$ achieves higher correctness than $V_{\text{SEQ}}$ for all $\mu < q/2 + 1/4$. With one additional batch, $V^2_{\text{GREEDY}}$ further improves the correctness of $V^1_{\text{GREEDY}}$ and outperforms $V_{\text{SEQ}}$ for all $\mu < q$.

$q \in \{0.6, 0.7, 0.8\}$, we study the behavior of $c(V^J_{\text{GREEDY}}), J \in \{1, 2\}$, as a function of the prior $\mu$. We observe several interesting properties. First, $\mu$ and $q$ apply two opposite forces on correctness: for both $J \in \{1, 2\}$, $c(V^J_{\text{OPT}})$ tends to decrease as $\mu$ increases but for higher $q$'s this effect is smaller. Intuitively, as the prior $\mu$ grows, the batch size becomes smaller (see Figure 3). As the planner learns from fewer signals, the probability to misallocate the object increases. However, as the signal precision $q$ improves, the magnitude of a misallocation decreases since signals become more reliable.

Second, the comparison between $V^1_{\text{GREEDY}}$ and $V^2_{\text{GREEDY}}$ confirms that an additional batch has an important positive effect on correctness. The two-batch mechanism $V^2_{\text{GREEDY}}$ outperforms the single-batch mechanism $V^1_{\text{GREEDY}}$. In fact, the gap between their achieved correctness grows as $\mu$ increases. Intuitively, if the planner fails to allocate a good object in the first batch, she has another chance to allocate it in the second batch. Since a higher $\mu$ translates to a higher probability that the object is of good quality, having two opportunities to allocate the object has a positive impact on correctness. This also explains the non-monotonic (but still generally decreasing) trend of $c(V^2_{\text{GREEDY}})$ with respect to $\mu$, which is especially evident for high values of $\mu$.

Finally, we are interested in comparing the correctness of voting mechanisms against two natural benchmarks. The first benchmark $V_{\text{SEQ}}$ is the sequential offering mechanism (see Section 3) in the presence of strategic incentives. As established in Theorem 1, there exists a voting mechanism (in this case, $V^2_{\text{GREEDY}}$) which outperforms $V_{\text{SEQ}}$ for all $\mu < q$. However, observe that the number of batches is crucial. In contrast to $V^2_{\text{GREEDY}}$, $V^1_{\text{GREEDY}}$ achieves higher correctness than $V_{\text{SEQ}}$ only for $\mu < \frac{q}{2} + \frac{1}{4}$.

Comparing $V^2_{\text{GREEDY}}$ against $V_{\text{SEQ}}$, we observe that the performance gap $c(V^2_{\text{GREEDY}}) - c(V_{\text{SEQ}})$ widens in the region $\mu \in [0, 1 - q]$ as $\mu$ grows. This is because the object is always discarded for $\mu < q$ (see Figure 1) which further explains why $c(V_{\text{SEQ}}) = 1 - \mu$ (Proposition 1). For $\mu \in (1 - q, q)$, the gap remains positive but decreases as $\mu$ increases, since the signals of the first two agents affect the outcome of $V_{\text{SEQ}}$ and thus decrease the chance of misallocation. Note again that, due to Theorem 1, for $\mu \geq q$, any voting mechanism $V \in \mathcal{V}$ achieves correctness equal to $c(V_{\text{SEQ}})$.

The second benchmark $V_{\text{ALL}}$ assumes that agents are not strategic (see Equation (2)) and thus achieves the optimal correctness close to 1. The ratio $c(V_{\text{GREEDY}}^J)/c(V_{\text{SEQ}})$, $J \in \{1, 2\}$, serves as a measure of the *price of anarchy*. We see that incentives put a significant cost on correctness, especially for higher values of $\mu \leq q$. Higher values of $q$ help decrease the price of anarchy. For any $q$, the maximum price of anarchy $c(V_{\text{GREEDY}}^J)/c(V_{\text{SEQ}})$ is observed around $\mu = q$, and equals $1/q$ in a single-batch mechanism.

## 6 Discussion and Conclusion

Recent empirical evidence suggests that a significant contributing factor to the high discard rate of donated organs is herding. We propose to amend the current allocation policy by incorporating learning and randomness into the allocation process. We support our recommendations by introducing and analyzing a stylized model through theory and simulations. From a theoretical perspective, our work highlights the tension between learning and incentives and contributes to the understanding of optimal allocation when agents are privately informed and thus susceptible to herding behavior.

**Limitations and extensions.** For simplicity, we make two limiting assumptions: (1) agent utilities are symmetric; (2) agents differ only in the realization of their private signals. We believe that our results are qualitatively robust and can be extended to a model with variations in agent types and asymmetric utilities. Increasing the relative magnitude of accepting a "bad" object will make agents more cautious, thus leading to larger batches. Introducing heterogeneous agents adds little insight to our qualitative conclusions; therefore, for the sake of brevity, we considered homogeneous agents.

Nevertheless, we acknowledge that our model is stylized with several abstractions that may not fully reflect the real organ allocation markets. In practice, an organ allocation waitlist might have patients of varying medical characteristics, such as age and blood type, which can give rise to heterogeneous utilities from the same organ. Therefore, our current model, which assumes a common utility function, should be interpreted as restricting attention to patients within the same medical characteristics group. It would be an interesting future direction to generalize some of these parts of our model.

**Social impact and ethical concerns.** Our proposed mechanism is designed to increase the acceptance rate of transplantable donated organs. While designing healthcare policies using insights from theoretical models has been extensively used (see e.g., [4, 5] for kidney exchange), we emphasize that one must pursue such a change in a gradual manner. Furthermore, we note that our mechanism does not constitute a Pareto improvement. Patients, who are first in line, may be skipped due to the randomness in allocation. Thus, although our policy has the potential to improve the overall system efficiency, its implementation must consider any unintended effects on individual patients.

## Acknowledgments and Disclosure of Funding

Jamie Kang acknowledges support of the Jerome Kaseberg Doolan Fellowship and Stanford Management Science & Engineering Graduate Fellowship.

Faidra Monachou acknowledges support from the Stanford Data Science Scholarship, Google Women Techmakers Scholarship, Stanford McCoy Family Center for Ethics in Society Graduate Fellowship, Jerome Kaseberg Doolan Fellowship, and A.G. Leventis Foundation Grant.

Moran Koren acknowledges the support of the Israeli Minestry of Science and Technologi's Shamir scholarship and the support of the Fulbright foundation.

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
