# A    Extended Related Literature

The information structure of the model is based on the literature on observational learning. The seminal papers by [7, 8] show that if agents receive binary signals over the object's quality and observe previous agents' action history, herding (information cascades) will eventually take place and information will be not be aggregated. These findings are extended by [28] to general distributions and follow-up works examine the robustness of these results in various game structures [1, 2, 14, 24]. Note that none of these works consider the existence of a planner that is able to control the flow of information.

Several empirical studies examine the presence of observational learning in real-world scenarios [13, 31]. Using data from organ transplantation in the United Kingdom, [13] develop empirical tests to detect herding behavior and quantify its welfare consequences. [31] provides empirical evidence for herding behavior in the United States deceased donor waitlist. Moreover, there also have been growing clinical literature on behavioral factors in organ allocation such as [9, 11].

To improve the performance of the allocation system, we propose utilizing a voting mechanism to elicit agents private information. The ability of voting procedures to uncover ground truth by aggregating dispersed information dates back to the seminal Condorcet Jury Theorem [12], which utilizes the law of large numbers to assert that majority voting will reach the correct decision, provided that the population is large enough and agents are not strategic. [6] show that Condorcet's result does not hold when agents are strategic and are allowed to deviate from their private information. Our notion of correctness is taken from [3]. One novel feature in our model is that the number of votes is determined endogenously to maximize the probability of making the correct choice.

Our work is also loosely connected to the literature on information design (e.g., [10, 18, 19, 26, 27]) and Bayesian exploration (e.g., [15–17, 20, 22]). With the exception of [15], who consider an online recommendation problem with costly information acquisition, most of these works do not take into account the possibility that agents are privately informed. Instead, several papers such as [17, 19] consider agents with private information in the form of *types* or idiosyncratic preferences. In contrast to our model where private *signals* are informative about the true quality of the object, those types are *not* correlated with the true state variable.[6]

# A    Optimal Solution in the Absence of Strategic Incentives

**Lemma A.1.** *Suppose that agents are not strategic and voluntarily reveal their private signal value to the planner. Let*

$$\underline{y} = 0.5 \, \frac{\log\left(\frac{1-\mu}{\mu}\right)}{\log\left(\frac{q}{1-q}\right)} + 0.5 \, I.$$

*Then, the planner achieves optimal correctness when she allocates the object if and only if $\sum_{i=1}^{I} \mathbf{1}\{s_i = g\} \geq \underline{y}$.*

*Proof.* Conditional on $\omega = G$, the number $X_G$ of positive signals $s = g$ follows a binomial distribution, i.e., $X_G \sim \text{Binomial}(I, q)$. Conditional on $\omega = G$, the number $X_B$ of positive signals $s = g$ follows a binomial distribution, i.e., $X_B \sim \text{Binomial}(I, 1 - q)$.

Let $\underline{y}$ denote the number of minimal positive signals required to allocate the object. For any number of positive signals higher than $\underline{y}$, it is also optimal to allocate the object since the posterior belief that $\omega = G$ strictly increases.

---

[6]I.e., in our case, knowing the value of private signals *does* lead the decision maker to update her belief accordingly to this information.

The threshold $\underline{y}$ is the smallest integer that satisfies

$$\mathbb{P}(\omega = G \mid X_G = \underline{y}) \geq 0.5$$

$$\iff \frac{\mu\binom{I}{\underline{y}}q^{\underline{y}}(1-q)^{I-\underline{y}}}{\mu\binom{I}{\underline{y}}q^{\underline{y}}(1-q)^{I-\underline{y}} + (1-\mu)\binom{I}{\underline{y}}q^{I-\underline{y}}(1-q)^{\underline{y}}} \geq 0.5$$

$$\iff \mu q^{\underline{y}}(1-q)^{I-\underline{y}} \geq (1-\mu)q^{I-\underline{y}}(1-q)^{\underline{y}}$$

$$\iff \mu q^{2\underline{y}-I} \geq (1-\mu)(1-q)^{2\underline{y}-I}$$

$$\iff \left(\frac{q}{1-q}\right)^{2\underline{y}-I} \geq \frac{1-\mu}{\mu}.$$

Recall that $q \in (0.5, 1)$. Taking the logarithm of the previous relation and rearranging the terms, we finally get that

$$\underline{y} = 0.5 \frac{\log\left(\frac{1-\mu}{\mu}\right)}{\log\left(\frac{q}{1-q}\right)} + 0.5\,I.$$

$\square$

# B    Omitted Analysis of the Sequential Offering Mechanism (Section 3)

We first analyze the optimal strategies of the first three agents in the queue. Before taking any action, each agent observes his position in the queue in addition to his private signal and the common object prior $\mu$. In particular, by observing his position $i$, agent $i$ recognizes that the object is being offered to him because all of the $i - 1$ preceding agents have rejected it. This observation contributes to his posterior belief about the object quality.

**Agent** 1. Agent 1 updates his posterior belief using Bayes' law based only on his signal $s_1$.

$$\mathbb{P}(\omega = G|s_1) = \begin{cases} \frac{\mu q}{\mu q + (1-\mu)(1-q)} & \text{for } s_1 = g \\ \frac{\mu(1-q)}{\mu(1-q)+(1-\mu)q} & \text{for } s_1 = b. \end{cases}$$

Since he receives utility 1 if $\omega = G$ and $-1$ if $\omega = B$, his expected utility to opt in is $\mathbb{P}(\omega = G|s_1) - (1 - \mathbb{P}(\omega = G|s_1))$. On the other hand, his utility to opt out is 0. We observe that agent 1 is better off opting in if

$$\mathbb{P}(\omega = G|s_1) - (1 - \mathbb{P}(\omega = G|s_1)) > 0,$$

which corresponds to

$$\begin{cases} \mu q - (1-\mu)(1-q) > 0 & \text{when } s_1 = g, \\ \mu(1-q) - (1-\mu)q > 0 & \text{when } s_1 = b. \end{cases}$$

Let $\alpha_1$ be agent 1's optimal action in this mechanism. Then the following lemma characterizes $\alpha_1$.

**Lemma B.1.** *Under the sequential offering mechanism, agent* 1 *chooses action*

$$\alpha_1 = \begin{cases} y & \text{for } \mu > q \\ \mathbb{I}\{s_1 = g\} & \text{for } \mu \in (1-q, q] \\ n & \text{for } \mu \leq 1-q. \end{cases}$$

Lemma B.1 shows that the sequential offering mechanism drives the first agent to follow his signal if it is more informative than the common prior (i.e., when $\mu \in (1 - q, q]$). In the opposite case, where the prior is more informative (i.e., when either $\mu > q$ or $\mu \leq 1 - q$), the agent ignores his private signal. In particular, for high priors $\mu > q$ the object is always allocated to agent 1 regardless of its true quality $\omega$.

**Agent** 2. When the second agent is offered the object, he knows that the first agent has already opted out. At the same time, agent 2 is aware of agent 1's optimal strategy as described in Lemma B.1.

Therefore, if the prior satisfies $\mu \in (1-q, q]$, then agent 2 infers that agent 1 has followed his own signal $s_1 = b$. As such, for $\mu \in (1-q, q]$, agent 2 updates his posterior belief informed by this observation as follows:

$$\mathbb{P}(\omega = G | s_2, \alpha_1 = n) = \begin{cases} \mu & \text{for } \mu \in (1-q, q], s_2 = g \\ \frac{\mu(1-q)^2}{\mu(1-q)^2 + (1-\mu)q^2} & \text{for } \mu \in (1-q, q], s_2 = b. \end{cases}$$

On the other hand, for $\mu \notin (1-q, q]$, it holds that

$$\mathbb{P}(\omega = G | s_2, \alpha_1 = n) = \begin{cases} \frac{\mu q}{\mu q + (1-\mu)(1-q)} & \text{for } \mu \notin (1-q, q], s_2 = g \\ \frac{\mu(1-q)}{\mu(1-q) + (1-\mu)q} & \text{for } \mu \notin (1-q, q], s_2 = b, \end{cases}$$

in which case agent 1's action is uninformative to agent 2.

**Lemma B.2.** *Under the sequential offering mechanism, agent 2 chooses*[7]

$$\alpha_2 = \begin{cases} \mathbb{I}\{s_2 = g\} & \text{for } \mu \in (1/2, q] \\ n & \text{for } \mu \leq 1/2. \end{cases}$$

*Proof.* Similarly to agent 1, agent 2 prefers to opt in if

$$\mathbb{P}(\omega = G | s_2, \alpha_1 = n) - (1 - \mathbb{P}(\omega = G | s_2, \alpha_1 = n)) > 0. \tag{3}$$

For $\mu \in (1-q, q]$ and $s_2 = g$, Equation (3) holds when $\mu > 1/2$. For $\mu \in (1-q, q]$ and $s_2 = b$, Equation (3) corresponds to

$$\mu(1-q)^2 - (1-\mu)q^2 > 0.$$

However, for any $\mu \in (1-q, q]$ we have

$$\mu(1-q)^2 - (1-\mu)q^2 \leq q(1-q)^2 - (1-q)q^2 = q(1-q)(1-2q) < 0,$$

which makes Equation (3) infeasible.

For $\mu \leq 1-q$, notice that

$$\mu(1-q) - (1-\mu)q < \mu q - (1-\mu)(1-q) \leq 0.$$

Thus Equation (3) is again infeasible regardless of $s_2$.

For $\mu = 1/2$ exactly, by our assumption, he simply rejects the object. Therefore, agent 2 prefers to opt in if $\mu \in (1/2, q]$ and $s_2 = g$. Otherwise, he prefers to opt out. $\square$

Lemma B.2 shows that agent 2 follows his signal if $\mu > 1/2$, whereas he ignores the signal and opts out if $\mu \leq 1/2$. The characterizations of $\alpha_1$ and $\alpha_2$ offer implications that will be useful to analyze the optimal strategies of subsequent agents. First, for $\mu > 1/2$, Lemma B.1 and Lemma B.2 together suggest that both agent 1 and agent 2 follow their signals. Therefore, the availability of the object after being offered to agent 2 implies to subsequent agents that both $s_1 = b$ and $s_2 = b$. Second, for $\mu \leq 1/2$, agent 2's action is uninformative about $s_2$. Agent 1's action, however, can be informative depending on the value of $\mu$. If $\mu \in (1-q, 1/2)$ then the subsequent agents can infer that $s_1 = b$ because agent 1 follows his signal. If $\mu \leq 1-q$, however, agent 1's action is also uninformative.

Notice here that, whenever agent 1's or 2's opt-out action is informative, his private signal (that the subsequent agents can perfectly infer) is $b$. We use this implication for our analysis next.

**Agent** 3. If the object is offered to agent 3, one of the three events must have occurred: (i) $\mu \in (1/2, q]$ *and* $s_1 = s_2 = b$, or (ii) $\mu \in (1-q, 1/2]$ *and* $s_1 = b$, or (iii) $\mu \leq 1-q$. Nonetheless, we claim that in any case, agent 3 would always choose to opt out.

**Lemma B.3.** *Under the sequential offering mechanism, agent 3 always chooses to opt out: $\alpha_3 = n$.*

---

[7]In the sequential offering mechanism, if $\mu > q$ then the object is never offered to agent $i \geq 2$, because agent 1 always opts in.

*Proof.* Agent 3 will prefer to opt in if

$$\mathbb{P}(\omega = G | s_3, \alpha_1 = \alpha_2 = n) - (1 - \mathbb{P}(\omega = G | s_3, \alpha_1 = \alpha_2 = n)) > 0. \tag{4}$$

For $\mu \in (1/2, q]$, agent 3's posterior belief given her signal is

$$\mathbb{P}(\omega = G | s_3, \alpha_1 = \alpha_2 = n) = \mathbb{P}(\omega = G | s_3, s_1 = s_2 = b)$$

$$= \begin{cases} \frac{\mu(1-q)^2 q}{\mu(1-q)^2 q + (1-\mu)q^2(1-q)} & \text{for } \mu \in (1/2, q], s_3 = g \\ \frac{\mu(1-q)^3}{\mu(1-q)^3 + (1-\mu)q^3} & \text{for } \mu \in (1/2, q], s_3 = b \end{cases}.$$

For $s_3 = g$, we have

$$\mu(1-q)^2 q - (1-\mu)q^2(1-q) = (1-q)q[\mu(1-q) - (1-\mu)q] \le 0.$$

The proof is analogous for $s_2 = b$. As a result, for $\mu \in (1/2, q]$ agent 3 prefers to opt out regardless of his own signal.

For $\mu \in (1-q, 1/2]$, agent 3 infers that $s_1 = b$. Agent 3's posterior belief equals

$$\mathbb{P}(\omega = G | s_3, \alpha_1 = \alpha_2 = n) = \mathbb{P}(\omega = G | s_3, s_1 = b)$$

$$= \begin{cases} \mu & \text{for } \mu \in (1-q, 1/2], s_3 = g \\ \frac{\mu(1-q)^2}{\mu(1-q)^2 + (1-\mu)q^2} & \text{for } \mu \in (1-q, 1/2], s_3 = b. \end{cases}$$

Then Equation (4) is infeasible because

$$\mu - (1-\mu) = 2\mu - 1 \le 0.$$

Finally, for $\mu \le 1 - q$,

$$\mathbb{P}(\omega = G | s_3, \alpha_1 = \alpha_2 = n) = \mathbb{P}(\omega = G | s_3)$$

$$= \begin{cases} \frac{\mu q}{\mu q + (1-\mu)(1-q)} & \text{for } \mu \le 1-q, s_3 = g \\ \frac{\mu(1-q)}{\mu(1-q) + (1-\mu)q} & \text{for } \mu \le 1-q, s_3 = b \end{cases}$$

where we have

$$\mu(1-q) - (1-\mu)q < \mu q - (1-\mu)(1-q) = \mu + q - 1 \le 0,$$

making Equation (4) infeasible.

$\square$

Now we extend Lemma B.3 to any remaining agent in the queue. Let $\alpha_i$ denote the optimal action of agent $i$.

**Lemma B.4.** *In the sequential offering mechanism, any agent other than the first two always opts-out. That is, for any $i \ge 3$ :*

$$\alpha_i = n.$$

*Proof.* We use mathematical induction on $i$ to prove the statement. For the basis $i = 3$, we have that due to Lemma B.3 agent 3 always opts out thus his action is always uninformative to the subsequent agents. Next, as the inductive step, consider some agent $i$. Suppose that agent $i$ updates his posterior belief in some manner such that it is optimal for him to opt out regardless of $s_i$. Because this action is uninformative, the next agent $i + 1$ must update his posterior belief the same way as agent $i$. Therefore, it is also optimal for agent $i + 1$ to opt out. $\square$

Another way to interpret Lemma B.4 is that if neither agent 1 nor agent 2 opts in, then the object will be discarded. Using this result along with Lemma B.1 and Lemma B.2 we characterize the outcome of the sequential offering mechanism (Lemma 1 in Section 3):

**Lemma 1.** *Under $V_{\text{SEQ}}$, the object is allocated if and only if: (i) $\mu > q$, (ii) $\mu \in [1/2, q]$ and either $s_1 = g$ or $s_2 = g$, (iii) $\mu \in [1-q, 1/2]$ and $s_1 = g$. Otherwise, the object is discarded.*

*Proof.* The proof follows directly from combining Lemmas B.1 to B.4. □

**Proposition 1.** *Under the sequential offering mechanism, $V_{\text{SEQ}}$:*

   *(i) At most the two initial agents in the queue determine the allocation outcome: if the first two agents decline, then the object will be always discarded.*

   *(ii) The correctness equals* $c(V_{\text{SEQ}}) = \begin{cases} \mu & \text{for } \mu > q \\ 2\mu q(1-q) + q^2 & \text{for } \mu \in [1/2, q] \\ q & \text{for } \mu \in [1-q, 1/2) \\ 1 - \mu & \text{for } \mu < 1 - q. \end{cases}$

*Proof.* Part (i) directly follows from Lemma 1. For part (ii), notice that for $\mu \in (1/2, q]$, the object is allocated if either $s_1 = g$ or $s_2 = g$. Conditional on $\omega = G$, this happens with probability $1 - (1-q)^2$. The object is discarded if $s_1 = s_2 = b$, which happens with $q^2$ conditional on $\omega = B$. Recall that $\mathbb{P}(\omega = G) = \mu$. Therefore the correctness of the outcome is $\mu\left(1 - (1-q)^2\right) + (1-\mu)q^2 = 2\mu q(1-q) + q^2$.

For $\mu \in (1-q, 1/2]$ the object is allocated if $s_1 = g$. Conditional on $\omega = G$, the object is allocated with probability $q$. The object is discarded when $s_1 = b$ and conditional on $\omega = B$ this happens with probability $q$. Hence the outcome is correct with probability $q$. □

## C   Proofs for Section 4

### C.1   Proofs for Section 4.1

**Lemma 2.** $V_{\{K\}}$ *is incentive-compatible if and only if (i) $\mu \in \mathcal{I}_K$, or equivalently (ii) $\mu < q$ and $\underline{K}(\mu) \leq K \leq \overline{K}(\mu)$.*

*Proof.* Proof of (i): Consider a single-batch voting mechanism $V_{\{K\}}$. Suppose that agent $i$ chooses to opt in. Let $\mathcal{G}_K$ be the probability that the object gets allocated to agent $i$ conditional on the object quality being good. Similarly, let $\mathcal{B}_K$ be the probability that the object gets allocated to agent $i$ conditional on the object quality being bad. Then $\mathcal{G}_K$ and $\mathcal{B}_K$ can be computed as

$$\mathcal{G}_K = \sum_{y=\frac{K+1}{2}}^{K} \frac{1}{y}\binom{K-1}{y-1} q^{y-1}(1-q)^{K-y}$$

$$\mathcal{B}_K = \sum_{y=\frac{K+1}{2}}^{K} \frac{1}{y}\binom{K-1}{y-1} q^{K-y}(1-q)^{y-1}.$$

Given private signal $s_i$, his expected utility conditional on each action is

$$u_i(y; s_i) = \mathbb{P}(\omega = G|s_i)\mathcal{G}_K - \mathbb{P}(\omega = B|s_i = g)\mathcal{B}_K$$
$$u_i(n; s_i) = 0.$$

Then the mechanism $V_{\{K\}}$ is incentive-compatible if $u_i(y; g) \geq u_i(n; g)$ and $u_i(y; b) \leq u_i(n; b)$. That is, $V_{\{K\}}$ is incentive-compatible for prior $\mu$ if

$$u_i(y; g) \geq u_i(n; g) \iff \mathbb{P}(\omega = G|s_i = g)\mathcal{G}_K - \mathbb{P}(\omega = B|s_i = g)\mathcal{B}_K \geq 0$$
$$\iff \frac{\mathbb{P}(\omega = G)\mathbb{P}(s_i = g|\omega = G)\mathcal{G}_K}{\mathbb{P}(s_i = g)} - \frac{\mathbb{P}(\omega = B)\mathbb{P}(s_i = g|\omega = B)\mathcal{B}_K}{\mathbb{P}(s_i = g)} \geq 0$$
$$\iff \mu q\mathcal{G}_K - (1-\mu)(1-q)\mathcal{B}_K \geq 0 \qquad (5)$$

and

$$u_i(y; b) \geq u_i(n; b) \iff \mathbb{P}(\omega = G|s_i = b)\mathcal{G}_K - \mathbb{P}(\omega = B|s_i = b)\mathcal{B}_K \leq 0$$
$$\iff \frac{\mathbb{P}(\omega = G)\mathbb{P}(s_i = b|\omega = G)\mathcal{G}_K}{\mathbb{P}(s_i = b)} - \frac{\mathbb{P}(\omega = B)\mathbb{P}(s_i = b|\omega = B)\mathcal{B}_K}{\mathbb{P}(s_i = b)} \leq 0$$
$$\iff \mu(1-q)\mathcal{G}_K - (1-\mu)q\mathcal{B}_K \leq 0. \qquad (6)$$

Given that the left-hand sides of both (5) and (6) monotonically increases with $\mu$, there should be some thresholds of prior, namely $\underline{\mu}_K \in (0,1)$ and $\overline{\mu}_K \in (0,1)$, such that

$$\mu q \mathcal{G}_K - (1-\mu)(1-q)\mathcal{B}_K \geq 0 \quad \forall \mu \geq \underline{\mu}_K$$

$$\mu(1-q)\mathcal{G}_K - (1-\mu)q\mathcal{B}_K \leq 0 \quad \forall \mu \geq \overline{\mu}_K,$$

where the thresholds solve the following indifference conditions:

$$\frac{\underline{\mu}_K}{1-\underline{\mu}_K} = \frac{1-q}{q}\frac{\mathcal{B}_K}{\mathcal{G}_K} \tag{7}$$

$$\frac{\overline{\mu}_K}{1-\overline{\mu}_K} = \frac{q}{1-q}\frac{\mathcal{B}_K}{\mathcal{G}_K}. \tag{8}$$

Because $q > \frac{1}{2}$ it naturally follows that $\underline{\mu}_K < \overline{\mu}_K$. Therefore, there exists a threshold policy such that any given $V_{\{K\}}$ is incentive-compatible for a prior $\mu$ that satisfies $\underline{\mu}_K \leq \mu \leq \overline{\mu}_K$.

Next we fully characterize $\underline{\mu}_K$ and $\overline{\mu}_K$. To do so, we use Lemma D.1 to write

$$\mathcal{G}_K = \frac{1}{q} \cdot \frac{1}{K} \sum_{y=\frac{K+1}{2}}^{K} \binom{K}{y} q^y (1-q)^{K-y}$$

$$\mathcal{B}_K = \frac{1}{1-q} \cdot \frac{1}{K} \sum_{y=0}^{\frac{K+1}{2}-1} \binom{K}{y} q^y (1-q)^{K-y}$$

and therefore

$$\frac{\mathcal{B}_K}{\mathcal{G}_K} = \frac{q}{1-q} \cdot \frac{1 - \mathbb{P}\left(X_K \geq \frac{K+1}{2}\right)}{\mathbb{P}\left(X_K \geq \frac{K+1}{2}\right)} \tag{9}$$

where $X_K$ is a binomial$(K,q)$ random variable. Plugging this into (7) and (8) we obtain

$$\underline{\mu}_K = 1 - \mathbb{P}\left(X \geq \frac{K+1}{2}\right) \tag{10}$$

$$\overline{\mu}_K = \frac{q^2\left(1 - \mathbb{P}\left(X \geq \frac{K+1}{2}\right)\right)}{q^2\left(1 - \mathbb{P}\left(X \geq \frac{K+1}{2}\right)\right) + (1-q)^2 \mathbb{P}\left(X \geq \frac{K+1}{2}\right)} \tag{11}$$

$$= \frac{q^2 \underline{\mu}_K}{q^2 \underline{\mu}_K + (1-q)^2(1-\underline{\mu}_K)}.$$

Letting $\mathcal{I}_K \triangleq \left(\underline{\mu}_K, \overline{\mu}_K\right)$ concludes the proof.

Proof of (ii): By Lemmas C.1 and C.2 (stated below), the set of $K$ such that $\mu \in \mathcal{I}_K$ must be comprised of consecutive odd numbers. Therefore, for any $K$ that satisfies $\underline{K}(\mu) \leq K \leq \overline{K}(\mu)$, the corresponding $V_{\{K\}}$ must be incentive-compatible. $\qquad\square$

We show two technical lemmas with regards to $\mathcal{I}_K$ that were used to prove Lemma 2. In doing so, we make use of Lemma D.2.

**Lemma C.1** (Decreasing $\mathcal{I}_K$). $\mathcal{I}_K$ *decreases with* $K$. *That is, the endpoints* $\overline{\mu}_K$ *and* $\underline{\mu}_K$ *both decrease with* $K$. *In particular,* $\overline{\mu}_1 = q$ *and* $\lim_{K\to\infty} \underline{\mu}_K = 0$.

*Proof.* Recall that the endpoints $\underline{\mu}_K$ and $\overline{\mu}_K$ are computed as Equations (10), both of which by Lemma D.2, decrease with $K$. To see the second part of the statement, since

$$\mathbb{P}(X_1 \geq 1) = \mathbb{P}(X_1 = 1) = q$$

it follows that $\overline{\mu}_1 = \frac{q^2(1-q)}{q^2(1-q)+(1-q)^2 q} = q$. Moreover,

$$\lim_{K\to\infty} \underline{\mu}_K = \lim_{K\to\infty} \frac{q^2\left(1 - \mathbb{P}\left(X_K \geq \frac{K+1}{2}\right)\right)}{q^2\left(1 - \mathbb{P}\left(X_K \geq \frac{K+1}{2}\right)\right) + (1-q)^2\mathbb{P}\left(X_K \geq \frac{K+1}{2}\right)} = 0$$

where the last equality is due to $\lim_{K\to\infty} \mathbb{P}\left(X_K \geq \frac{K+1}{2}\right) = 1$ by Lemma D.2. $\qquad\square$

**Lemma C.2** (Overlapping $\mathcal{I}_K$). *For any batch size $K \geq 3$, the interval $\mathcal{I}_K$ intersects with $\mathcal{I}_{K-2}$. That is,*

$$\underline{\mu}_K < \underline{\mu}_{K-2} < \overline{\mu}_K < \overline{\mu}_{K-2}.$$

*Proof. (Sketch)* The first and third inequalities are immediate from Lemma C.1. Showing the second inequality involves finding appropriate bounds for $\frac{\mathcal{B}_K}{\mathcal{G}_K} \frac{\mathcal{G}_{K-2}}{\mathcal{B}_{K-2}}$. Due to the length of the proof, we defer this part to Appendix C.4. □

Figure 2 illustrates Lemmas C.1 and C.2.

**Lemma 3.** *Suppose $\mu < q$. Then, an incentive-compatible $V_{\{K\}}$ always exists and achieves correctness $c\left(V_{\{K\}}\right) = \mathbb{P}\left(X_K \geq \frac{K+1}{2}\right)$, where $X_K \sim Binomial(K, q)$.*

*Proof.* We first show the existence of incentive-compatible $V_{\{K\}}$ for any $\mu < q$. Consider some $\mathcal{K}$. By Lemmas C.1 and C.2,

$$\bigcup_{K=1}^{\mathcal{K}} \mathcal{I}_K = (\underline{\mu}_{\mathcal{K}}, q)$$

which implies that that no value of prior $\mu$ between $\underline{\mu}_{\mathcal{K}}$ and $q$ is skipped. Using the asymptotic result from the second part of Lemma C.1,

$$\lim_{\mathcal{K} \to \infty} \bigcup_{K=1}^{\mathcal{K}} \mathcal{I}_K = \left(\lim_{\mathcal{K} \to \infty} \underline{\mu}_{\mathcal{K}}, q\right) = (0, q).$$

Therefore for any $\mu$ such that $\mu \leq q$, there exists at least one $K$ such that $\mu \in \mathcal{I}_K$.

For the second part, we have

$$\begin{aligned}
c\left(V_{\{K\}}\right) &= \mu \, \mathbb{P}_{V_{\{K\}}}(Z = 1 \mid \omega = G) + (1 - \mu) \, \mathbb{P}_{V_{\{K\}}}(Z = 0 \mid \omega = B) \\
&= \mu \mathbb{P}\left(Y \geq \frac{K+1}{2} \, \bigg| \, \omega = G\right) + (1 - \mu)\mathbb{P}\left(Y < \frac{K+1}{2} \, \bigg| \, \omega = B\right) \\
&= \mathbb{P}\left(X_K \geq \frac{K+1}{2}\right).
\end{aligned}$$

□

**Proposition 2.** *Suppose $\mu < q$. Then the batch size $K = \overline{K}(\mu)$ maximizes correctness among all incentive-compatible $V_{\{K\}}$.*

*Proof.* By Lemma 3, $c\left(V_{\{K\}}\right) = \mathbb{P}\left(X_K \geq \frac{K+1}{2}\right)$, which by Lemma D.2, increases with $K$. □

**Lemma 4.** $\overline{K}(\mu)$ *weakly decreases with $\mu$.*

*Proof.* See the proof of Lemma C.1. □

## C.2   Proofs for Section 4.2

**Lemma 5.** *A voting mechanism $V_{\{\pi_j\}_{j=1}^{\infty}}$ is incentive-compatible if and only if for any batch $j$, current belief $\mu_{j-1}$ and batch size $K_j \triangleq \pi_j(\mu_{j-1})$, $V_{\{K_j\}}$ is incentive-compatible.*

*Proof.* Each agent participates in at most one batch. Furthermore, agents in the same batch $j$ share the same belief $\mu_{j-1}$ with the planner. Thus, given the current belief $\mu_{j-1}$ and batch size $K_j$ (given the outcome of batch $j - 1$), the mechanism $V_{\{\pi_j\}_{j=1}^{\infty}}$ restricted to batch $j$ is equivalent to a single-batch mechanism $V_{\{K_j\}}$ with a current belief $\mu_{j-1}$. Therefore, by induction, the mechanism $V_{\{\pi_j\}_{j=1}^{\infty}}$ is incentive-compatible if and only if for each batch $j$, the single-batch mechanism $V_{\{K_j\}}$ with belief $\mu_{j-1}$ is incentive-compatible. □

**Lemma C.3.** *For $\mu \geq q$, there is no incentive-compatible voting mechanism; under any $V \in \mathcal{V}$, every agent opts in.*

*Proof.* By Lemma 5, it suffices to consider single-batch voting mechanisms. For any $K$, Lemma C.1 implies that

$$\overline{\mu}_K \leq \max_K \overline{\mu}_K = \overline{\mu}_1 = q.$$

Therefore for all $\mu \geq q$, there exists no $K$ such that $\mu < \overline{\mu}_K$. For these priors, every agent will choose (possibly untruthfully) $\alpha_i = y$ under any voting mechanism, and therefore the mechanism always allocates the object (and terminates after one batch). In expectation, such an outcome is correct with probability $\mu$. Notice that the correctness of this outcome is equivalent to that under $V_{\text{SEQ}}$ for $\mu \geq q$. $\qquad\square$

**Lemma C.4.** *Suppose that the object is offered to batch $j$ but is not allocated. Then $\mu_j < \mu_{j-1}$.*

*Proof.* Object is not allocated to batch $j$ if $Y_j \leq \frac{K_j - 1}{2}$. Using the posterior update rule in (1),

$$
\begin{aligned}
\mu_j &= \frac{\mu_{j-1} q^{Y_j} (1-q)^{K_j - Y_j}}{\mu_{j-1} q^{Y_j} (1-q)^{K_j - Y_j} + (1 - \mu_{j-1}) q^{K_j - Y_j} (1-q)^{Y_j}} \\
&\leq \frac{\mu_{j-1}(1-q)}{\mu_{j-1}(1-q) + (1 - \mu_{j-1})q} \\
&< \mu_{j-1}.
\end{aligned}
$$

$\qquad\square$

**Proposition 3.** *For any $\mu < q$ and $J$, $V_{\text{GREEDY}}^J$ has the following properties:*
   *(i) $V_{\text{GREEDY}}^J$ is incentive-compatible;*
   *(ii) Ex-post batch sizes satisfy $K_{j'} \leq K_j$ for any $j' < j \in [J]$;*
   *(iii) $c\left(V_{\text{GREEDY}}^J\right)$ strictly increases with $J$.*

*Proof.* Part (i) is by Lemma 5.

For part (ii), by Lemma C.4, $\mu_j$ decreases with $j$. Each batch size $K_j$ is chosen as $K_j = \overline{K}(\mu_{j-1})$, which, by Lemma 4, weakly increases with $j$.

For part (iii), consider any integer $J$ and the greedy mechanism $V_{\text{GREEDY}}^J$. Suppose that the object has not been allocated up to batch $J$. Then should the planner offer the object to another batch?

On the one hand, the expected correctness conditional on stopping at batch $J$ is

$$\mathbb{P}\left(\omega = B \mid \mu_J\right) = 1 - \mu_J.$$

On the other hand, the expected correctness conditional on offering to an additional batch $J+1$ is equal to $c\left(V_{\{K_{J+1}\}}\right)$, which is the correctness of a single-batch voting mechanism where $K_{J+1} = \overline{K}(\mu_J)$ is the size of batch $J+1$ chosen in a greedy manner. Then by Lemma 3, the expected correctness is

$$\mathbb{P}\left(X_{\overline{K}(\mu_J)} \geq \frac{\overline{K}(\mu_J) + 1}{2}\right).$$

Consider the interval $\mathcal{I}_{\overline{K}(\mu_J)}$ that defines the values of incentive-compatible priors for the single-batch voting mechanism $V_{\{\overline{K}(\mu_J)\}}$. In particular, consider the lower endpoint of this interval, $\underline{\mu}_{\overline{K}(\mu_J)}$. Then, because $\mu_J \in \mathcal{I}_{\overline{K}(\mu_J)}$ by construction, it must be that $\mu_J > \underline{\mu}_{\overline{K}(\mu_J)}$. Furthermore, we can compute

$$\underline{\mu}_{\overline{K}(\mu_J)} = 1 - \mathbb{P}\left(X_{\overline{K}(\mu_J)} \geq \frac{\overline{K}(\mu_J) + 1}{2}\right).$$

Hence, the expected correctness achieved by an additional batch is

$$
\begin{aligned}
c\left(V_{\{\overline{K}(\mu_J)\}}\right) &= \mathbb{P}\left(X_{\overline{K}(\mu_J)} \geq \frac{\overline{K}(\mu_J) + 1}{2}\right) \\
&= 1 - \underline{\mu}_{\overline{K}(\mu_J)} \\
&> 1 - \mu_J.
\end{aligned}
$$

$\qquad\square$

**Corollary 1.** *For any $\mu < q$, $V_{\mathrm{GREEDY}}$ is an incentive-compatible (multi-batch) voting mechanism that improves correctness in comparison to the sequential offering mechanism $V_{\mathrm{SEQ}}$.*

*Proof.* Immediate from Theorem 1 and Proposition 3 (part (iii)). $\square$

### C.3 Proof of Theorem 1

**Theorem 1.** *For any $\mu < q$, there exists a voting mechanism $V \in \mathcal{V}$ that is incentive-compatible and improves correctness compared to the sequential offering mechanism $V_{\mathrm{SEQ}}$. For $\mu \geq q$, there is no incentive-compatible voting mechanism and any $V \in \mathcal{V}$ achieves the same correctness as $V_{\mathrm{SEQ}}$.*

*Proof.* In Proposition 1 we calculate the correctness of $V_{SEQ}$, which will be used as our benchmark. We split the proof into three different cases:

- Case I: $\mu < 1 - q$, so by Proposition 1,

$$c(V_{\mathrm{SEQ}}) = 1 - \mu.$$

  By Lemmas 3, there must exist an incentive-compatible single-batch voting mechanism $V_{\{K\}}$ such that $\mu \in \mathcal{I}_K$. In particular, under such a mechanism,

$$\mu > 1 - \mathbb{P}\left(K \geq \frac{K+1}{2}\right) = 1 - c\left(V_{\{K\}}\right).$$

  Therefore,

$$c(V_{\{K\}}) > 1 - \mu = c(V_{\mathrm{SEQ}}).$$

- Case II: $\mu \in [1 - q, 1/2)$.

$$c(V_{\mathrm{SEQ}}) = q,$$

  and more importantly, for these values of prior, $V_{\mathrm{SEQ}}$ is equivalent to $V_{\{1\}}$ that is incentive-compatible. Therefore, we use the observation (by Lemma 3 and D.2) that the correctness of an incentive-compatible $V_{\{K\}}$ increases with $K$.

  Then it is sufficient to show that there exists an incentive-compatible $V_{\{K\}}$ such that $K \geq 3$. To this end, we show that $V_{\{3\}}$ is incentive-compatible. The interval $\mathcal{I}_3$ computes to

$$\mathcal{I}_3 = \left((1-q)^2(2q+1), \frac{q}{2} + \frac{1}{4}\right)$$

  where $(1-q)^2(2q+1) < 1 - q$ and $\frac{q}{2} + \frac{1}{4} > \frac{1}{2}$. Hence

$$[1 - q, 1/2) \subset \mathcal{I}_3,$$

  implying that $V_{\{3\}}$ is incentive-compatible for any $\mu \in [1 - q, 1/2)$.

- Case III: $\mu \in [1/2, q/2 + 1/4]$, in which case $\frac{1}{2} \leq \mu < q$. By Proposition 1,

$$c(V_{\mathrm{SEQ}}) = 2\mu q(1 - q) + q^2.$$

  As in Case II,

$$[1/2, q/2 + 1/4] \subset \mathcal{I}_3,$$

  and therefore $V_{\{3\}}$ is incentive-compatible, which achieves (by Lemma 3)

$$c\left(V_{\{3\}}\right) = q^3 + 3q^2(1 - q) > 2\mu q(1 - q) + q^2 = c(V_{\mathrm{SEQ}}).$$

- Case IV: $\mu \in (q/2 + 1/4, q)$.

$$c(V_{\mathrm{SEQ}}) = 2\mu q(1 - q) + q^2.$$

  Notice that $V_{\{3\}}$ is no longer incentive-compatible for these values of $\mu$. Since $\mathcal{I}_1 = (1 - q, q)$, $V_{\{1\}}$ is the only incentive-compatible single-batch voting mechanism, which however achieves $c\left(V_{\{1\}}\right) = q < c(V_{\mathrm{SEQ}})$.

Motivated by Proposition 3 (especially, part (iii)), we turn to voting mechanisms with multiple batches instead. Consider $V_{\text{GREEDY}}^2$. Since $V_{\{1\}}$ is the only incentive-compatible single-batch voting mechanism, in order for $V_{\text{GREEDY}}^2$ to be incentive-compatible, it must have $K_1 = 1$. The correctness of this mechanism is then

$$
\begin{aligned}
c(V_{\text{GREEDY}}^2) &= \mathbb{E}_{V_{\text{GREEDY}}^2}\left[\mathbb{I}(\omega = G \cap Z = 1) + \mathbb{I}(\omega = B \cap Z = 0)\right] \\
&= \mathbb{P}\left(Y_1 = 1 \cap \omega = G \mid K_1 = 1\right) \\
&\quad + \mathbb{P}\left(Y_1 = 0 \mid K_1 = 1\right) \max_{K_2} \mathbb{E}\left[c(V_{\{K_2\}}) \mid Y_1 = 0, K_1 = 0\right] \\
&= \mu q + (\mu(1-q) + (1-\mu)q)\, \mathbb{E}\left[c(V_{\{\overline{K}(\mu_1)\}}) \mid \mu_1 = \frac{\mu(1-q)}{\mu(1-q) + (1-\mu)q}\right].
\end{aligned}
$$

Here, in the event where the object is not allocated to the first batch (i.e., $Y_1 = 0$),

$$
\mu_1 \in \left(\frac{\left(\frac{q}{2} + \frac{1}{4}\right)(1-q)}{\left(\frac{q}{2} + \frac{1}{4}\right)(1-q) + \left(1 - \frac{q}{2} - \frac{1}{4}\right)q}, \frac{1}{2}\right)
$$

each of which is achieved by plugging in $\mu = \frac{q}{2} + \frac{1}{4}$ and $\mu = q$ respectively. In particular, observe that

$$
\mu_1 < \frac{1}{2} < \frac{q}{2} + \frac{1}{4} = \overline{\mu}_3,
$$

where $\overline{\mu}_3$ denotes the upper endpoint of the interval $\mathcal{I}_3$. This in turn implies

$$
\overline{K}(\mu_1) \geq \overline{K}(\overline{\mu}_3) = 3
$$

where the inequality is by Lemma 4. Therefore,

$$
\mathbb{E}\left[c(V_{\{\overline{K}(\mu_1)\}}) \mid \mu_1 = \frac{\mu(1-q)}{\mu(1-q) + (1-\mu)q}\right] \geq c(V_{\{3\}}) = q^3 + 3q^2(1-q).
$$

Using this observation,

$$
c(V_{\text{GREEDY}}^2) \geq \mu q + (\mu(1-q) + (1-\mu)q)\left(q^3 + 3q^2(1-q)\right)
$$

and more importantly,

$$
c(V_{\text{GREEDY}}^2) - c(V_{\text{SEQ}}^2) \geq \mu q + (\mu(1-q) + (1-\mu)q)\left(q^3 + 3q^2(1-q)\right) - 2\mu q(1-q) - q^2.
$$

The partial derivative (with respect to $\mu$) of the inequality's right-hand side is negative. Therefore, plugging in $\mu = q$ yields

$$
c(V_{\text{GREEDY}}^2) - c(V_{\text{SEQ}}^2) > q^2 + 2q(1-q)\left(q^3 + 3q^2(1-q)\right) - 2q^2(1-q) - q^2
$$

where the right-hand side term is always positive for any $q \in (0.5, 1)$. As a result,

$$
c(V_{\text{GREEDY}}^2) > c(V_{\text{SEQ}}).
$$

Notice in Cases II and III, we use $V_{\{3\}}$ as a sufficient condition. However, one can always further improve correctness by using either the optimal $V_{\text{GREEDY}}^1$ for the given prior or any other $V_{\text{GREEDY}}^J$ (including $V_{\text{GREEDY}}$ implemented as Algorithm 1) with more batches.

Finally, the proof of the second statement is given in Lemma C.3.

$\square$

## C.4 Remaining Proof of Lemma C.2

**Lemma C.2** (Overlapping $\mathcal{I}_K$). *For any batch size $K \geq 3$, the interval $\mathcal{I}_K$ intersects with $\mathcal{I}_{K-2}$. That is,*

$$
\underline{\mu}_K < \underline{\mu}_{K-2} < \overline{\mu}_K < \overline{\mu}_{K-2}.
$$

*Proof.* First and third inequalities are immediate from Lemma C.1. Hence our focus is to show that for any $K \geq 3$,

$$\underline{\mu}_{K-2} \leq \overline{\mu}_K.$$

By the indifference conditions (7) and (8), we can instead show $\frac{1-q}{q} \frac{\mathcal{B}_{K-2}}{\mathcal{G}_{K-2}} < \frac{q}{1-q} \frac{\mathcal{B}_K}{\mathcal{G}_K}$, or equivalently

$$\left(\frac{1-q}{q}\right)^2 < \frac{\mathcal{B}_K}{\mathcal{G}_K} \frac{\mathcal{G}_{K-2}}{\mathcal{B}_{K-2}},$$

where $\mathcal{G}_K$ (respectively, $\mathcal{B}_K$) are defined in Section 4.1 as the probability that the object gets allocated to some agent in the batch of size $K$ conditional on the true quality of the object being good (respectively, bad). We show the last inequality in three steps. Each step involves finding a tighter lower bound for $\frac{\mathcal{B}_K}{\mathcal{G}_K} \frac{\mathcal{G}_{K-2}}{\mathcal{B}_{K-2}}$ than the previous step.

First of all, we bound $\frac{\mathcal{B}_K}{\mathcal{G}_K} \frac{\mathcal{G}_{K-2}}{\mathcal{B}_{K-2}}$ using a function of binomial distributions.

Recall the indifference conditions (7) and (8):

$$\frac{\underline{\mu}_K}{1-\underline{\mu}_K} = \frac{1-q}{q} \frac{\mathcal{B}_K}{\mathcal{G}_K}, \quad \frac{\overline{\mu}_K}{1-\overline{\mu}_K} = \frac{q}{1-q} \frac{\mathcal{B}_K}{\mathcal{G}_K}.$$

Rearranging the terms yields

$$\frac{\mathcal{B}_K}{\mathcal{G}_K} = \frac{\underline{\mu}_K}{1-\underline{\mu}_K} \frac{q}{1-q} = \frac{1-\mathbb{P}\left(X_K \geq \frac{K+1}{2}\right)}{\mathbb{P}\left(X_K \geq \frac{K+1}{2}\right)} \frac{q}{1-q}$$

$$\frac{\mathcal{G}_{K-2}}{\mathcal{B}_{K-2}} = \frac{1-\underline{\mu}_{K-2}}{\underline{\mu}_{K-2}} \frac{1-q}{q} = \frac{\mathbb{P}\left(X_{K-2} \geq \frac{K+1}{2} - 2\right)}{1-\mathbb{P}\left(X_{K-2} \geq \frac{K+1}{2} - 2\right)} \frac{1-q}{q}.$$

Therefore,

$$\frac{\mathcal{B}_K}{\mathcal{G}_K} \frac{\mathcal{G}_{K-2}}{\mathcal{B}_{K-2}} = \frac{1-\mathbb{P}\left(X \geq \frac{K+1}{2}\right)}{\mathbb{P}\left(X \geq \frac{K+1}{2}\right)} \frac{\mathbb{P}\left(X_{K-2} \geq \frac{K+1}{2} - 2\right)}{1-\mathbb{P}\left(X_{K-2} \geq \frac{K+1}{2} - 2\right)}$$

$$= \frac{\sum_{y=0}^{\frac{K+1}{2}-1} \binom{K}{y} q^y (1-q)^{K-y}}{\sum_{y=0}^{\frac{K+1}{2}-1} \binom{K}{y} (1-q)^y q^{K-y}} \frac{\sum_{y=0}^{\frac{K+1}{2}-2} \binom{K-2}{y} (1-q)^y q^{K-2-y}}{\sum_{y=0}^{\frac{K+1}{2}-2} \binom{K-2}{y} q^y (1-q)^{K-2-y}}.$$

Observe that the last expression can be written as

$$\left(\frac{\sum_{y=0}^{\frac{K+1}{2}-2} \binom{K}{y} q^y (1-q)^{K-y} + \binom{K}{\frac{K+1}{2}-1} q^{\frac{K+1}{2}-1} (1-q)^{\frac{K+1}{2}}}{\sum_{y=0}^{\frac{K+1}{2}-2} \binom{K}{y} (1-q)^y q^{K-y} + \binom{K}{\frac{K+1}{2}-1} (1-q)^{\frac{K+1}{2}-1} q^{\frac{K+1}{2}}}\right) \left(\frac{\sum_{y=0}^{\frac{K+1}{2}-2} \binom{K-2}{y} (1-q)^y q^{K-2-y}}{\sum_{y=0}^{\frac{K+1}{2}-2} \binom{K-2}{y} q^y (1-q)^{K-2-y}}\right)$$

which is lower bounded by (by Lemma D.4)

$$\frac{\sum_{y=0}^{\frac{K+1}{2}-2} \binom{K}{y} q^y (1-q)^{K-y}}{\sum_{y=0}^{\frac{K+1}{2}-2} \binom{K}{y} (1-q)^y q^{K-y}} \frac{\sum_{y=0}^{\frac{K+1}{2}-2} \binom{K-2}{y} (1-q)^y q^{K-2-y}}{\sum_{y=0}^{\frac{K+1}{2}-2} \binom{K-2}{y} q^y (1-q)^{K-2-y}}.$$

Hence

$$\frac{\mathcal{B}_K}{\mathcal{G}_K} > \frac{\sum_{y=0}^{\frac{K+1}{2}-2} \binom{K}{y} q^y (1-q)^{K-y}}{\sum_{y=0}^{\frac{K+1}{2}-2} \binom{K}{y} (1-q)^y q^{K-y}} \frac{\sum_{y=0}^{\frac{K+1}{2}-2} \binom{K-2}{y} (1-q)^y q^{K-2-y}}{\sum_{y=0}^{\frac{K+1}{2}-2} \binom{K-2}{y} q^y (1-q)^{K-2-y}}. \tag{12}$$

The second step involves rearranging the lower bound in (12) found in the previous step. We introduce the following notations.

$$\eta_q \triangleq \sum_{y=0}^{\frac{K+1}{2}-2} \binom{K}{y} q^y (1-q)^{K-y} \binom{K-2}{y} (1-q)^y q^{K-2-y}$$

$$\eta_{1-q} \triangleq \sum_{y=0}^{\frac{K+1}{2}-2} \binom{K}{y} (1-q)^y q^{K-y} \binom{K-2}{y} q^y (1-q)^{K-2-y}$$

$$\psi_q \triangleq \sum_{y=0}^{\frac{K+1}{2}-2} \sum_{z=y+1}^{\frac{K+1}{2}-2} \left[ \binom{K}{y} q^y (1-q)^{K-y} \binom{K-2}{z} (1-q)^z q^{K-2-z} \right.$$

$$\left. + \binom{K}{z} q^z (1-q)^{K-z} \binom{K-2}{y} (1-q)^y q^{K-2-y} \right]$$

$$\psi_{1-q} \triangleq \sum_{y=0}^{\frac{K+1}{2}-2} \sum_{z=y+1}^{\frac{K+1}{2}-2} \left[ \binom{K}{y} (1-q)^y q^{K-y} \binom{K-2}{z} q^z (1-q)^{K-2-z} \right.$$

$$\left. + \binom{K}{z} (1-q)^z q^{K-z} \binom{K-2}{y} q^y (1-q)^{K-2-y} \right].$$

Then the right-hand side term of (12) is equal to $\frac{\eta_q + \psi_q}{\eta_{1-q} + \psi_{1-q}}$, and thus

$$\frac{\mathcal{B}_K}{\mathcal{G}_K} \frac{\mathcal{G}_{K-2}}{\mathcal{B}_{K-2}} = \frac{\eta_q + \psi_q}{\eta_{1-q} + \psi_{1-q}}.$$

Since $\eta_q + \psi_q$ and $\eta_{1-q} + \psi_{1-q}$ are series of the same length $\frac{K^2-1}{8} \left( = \frac{(\frac{K+1}{2}-1)\frac{K+1}{2}}{2} \right)$, we use Lemma D.3 twice to find the following lower bound.

$$\frac{\eta_q + \psi_q}{\eta_{1-q} + \psi_{1-q}} \geq \min\left\{ \frac{\eta_q}{\eta_{1-q}}, \frac{\psi_q}{\psi_{1-q}} \right\}$$

$$\geq \min\left\{ \min_{y \in \{0,\dots,\frac{K+1}{2}-2\}} \frac{\binom{K}{y}\binom{K-2}{y} q^{K-2}(1-q)^K}{\binom{K}{y}\binom{K-2}{y}(1-q)^{K-2}q^K}, \right.$$

$$\left. \min_{\substack{(y,z)|y\in\{0,\dots,\frac{K+1}{2}-2\},\\ z\in\{y+1,\dots,\frac{K+1}{2}-2\}}} \frac{\binom{K}{y}\binom{K-2}{z} q^{K-2+y-z}(1-q)^{K-y+z} + \binom{K}{z}\binom{K-2}{y} q^{K-2-y+z}(1-q)^{K+y-z}}{\binom{K}{y}\binom{K-2}{z}(1-q)^{K-2+y-z}q^{K-y+z} + \binom{K}{z}\binom{K-2}{y}(1-q)^{K-2-y+z}q^{K+y-z}} \right\}$$

where both inequalities follow by Lemma D.3.

The final step involves finding the lower bounds of these terms. For any $y$ we have

$$\frac{\binom{K}{y}\binom{K-2}{y} q^{K-2}(1-q)^K}{\binom{K}{y}\binom{K-2}{y}(1-q)^{K-2}q^K} = \left(\frac{1-q}{q}\right)^2$$

and thus

$$\min_{y \in \{0,\dots,\frac{K+1}{2}-2\}} \frac{\binom{K}{y}\binom{K-2}{y} q^{K-2}(1-q)^K}{\binom{K}{y}\binom{K-2}{y}(1-q)^{K-2}q^K} = \left(\frac{1-q}{q}\right)^2.$$

Similarly for any $y \in \{0, \ldots, \frac{K+1}{2} - 2\}, z \in \{y+1, \ldots, \frac{K+1}{2} - 2\}$,

$$\frac{\binom{K}{y}\binom{K-2}{z}q^{K-2+y-z}(1-q)^{K-y+z} + \binom{K}{z}\binom{K-2}{y}q^{K-2-y+z}(1-q)^{K+y-z}}{\binom{K}{y}\binom{K-2}{z}(1-q)^{K-2+y-z}q^{K-y+z} + \binom{K}{z}\binom{K-2}{y}(1-q)^{K-2-y+z}q^{K+y-z}}$$

$$= \frac{q^{K-2+y-z}(1-q)^{K+y-z}\left[\binom{K}{y}\binom{K-2}{z}(1-q)^{2(z-y)} + \binom{K}{z}\binom{K-2}{y}q^{2(z-y)}\right]}{(1-q)^{K-2+y-z}q^{K+y-z}\left[\binom{K}{y}\binom{K-2}{z}q^{2(z-y)} + \binom{K}{z}\binom{K-2}{y}(1-q)^{2(z-y)}\right]}$$

$$= \left(\frac{1-q}{q}\right)^2 \frac{\binom{K}{y}\binom{K-2}{z}(1-q)^{2(z-y)} + \binom{K}{z}\binom{K-2}{y}q^{2(z-y)}}{\binom{K}{y}\binom{K-2}{z}q^{2(z-y)} + \binom{K}{z}\binom{K-2}{y}(1-q)^{2(z-y)}}$$

$$> \left(\frac{1-q}{q}\right)^2$$

To show the last inequality above, because $z > y$ and $q > 1 - q$, it is sufficient to show that

$$\binom{K}{z}\binom{K-2}{y} > \binom{K}{y}\binom{K-2}{z}. \tag{13}$$

Observe that

$$\binom{K}{z}\binom{K-2}{y} = \frac{K!}{z!(K-z)!}\frac{(K-2)!}{y!(K-y-2)!}$$

$$\binom{K}{y}\binom{K-2}{z} = \frac{K!}{y!(K-y)!}\frac{(K-2)!}{z!(K-z-2)!}.$$

Then (13) holds since

$$\frac{1}{(K-z)!(K-y-2)!} > \frac{1}{(K-y)!(K-z-2)!}$$

because

$$(K-y-1)(K-y) = \frac{(K-y)!}{(K-y-2)!} > \frac{(K-z)!}{(K-z-2)!} = (K-z-1)(K-z).$$

Therefore this concludes the proof of Lemma C.2:

$$\frac{\mathcal{B}_K}{\mathcal{G}_K}\frac{\mathcal{G}_{K-2}}{\mathcal{B}_{K-2}} > \frac{\eta_q + \psi_q}{\eta_{1-q} + \psi_{1-q}} > \left(\frac{1-q}{q}\right)^2$$

where the first lower bound was achieved via the first step, and the final tighter lower bound was achieved via the second and third steps. $\qquad\square$

## D   Technical Properties

**Lemma D.1.** $\frac{1}{y}\binom{K-1}{y-1} = \frac{1}{K}\binom{K}{y}$.

*Proof.*

$$\frac{1}{y}\binom{K-1}{y-1} = \frac{1}{y}\frac{(K-1)!}{(y-1)!(K-y)!}$$

$$= \frac{(K-1)!}{y!(K-y)!}$$

$$= \frac{1}{K}\frac{K!}{y!(K-y)!}$$

$$= \frac{1}{K}\binom{K}{y}.$$

$\qquad\square$

**Lemma D.2.** *For any $q \in (0.5, 1)$ and odd $K \in \mathbb{N}$, let $X_K$ be a Binomial$(K, q)$ random variable.*

$$\mathbb{P}\left(X_{K+2} \geq \frac{(K+2)+1}{2}\right) > \mathbb{P}\left(X_K \geq \frac{K+1}{2}\right)$$

*and as $K$ grows*

$$\lim_{K \to \infty} \mathbb{P}\left(X_K \geq \frac{K+1}{2}\right) = 1.$$

*Proof.* To show the first part, we use the following property of the binomial distribution

$$\mathbb{P}\left(X_{K+1} \geq \frac{K+1}{2}+1\right) = \mathbb{P}\left(X_K \geq \frac{K+1}{2}+1\right) + q\mathbb{P}\left(X_K \geq \frac{K+1}{2}\right).$$

Then in follows that

$$\mathbb{P}\left(X_{K+2} \geq \frac{(K+2)+1}{2}\right) = \mathbb{P}\left(X_{K+1} \geq \frac{K+1}{2}+1\right) + q\mathbb{P}\left(X_{K+1} \geq \frac{K+1}{2}\right)$$

$$= \mathbb{P}\left(X_K \geq \frac{K+1}{2}+1\right) + q\mathbb{P}\left(X_K \geq \frac{K+1}{2}\right)$$

$$+ q\left[\mathbb{P}\left(X_K \geq \frac{K+1}{2}\right) + q\mathbb{P}\left(X_K \geq \frac{K+1}{2}-1\right)\right]$$

$$= 2q\mathbb{P}\left(X_K \geq \frac{K+1}{2}\right) + \mathbb{P}\left(X_K \geq \frac{K+1}{2}+1\right) + q^2\mathbb{P}\left(X_K \geq \frac{K+1}{2}-1\right)$$

$$> \mathbb{P}\left(X_K \geq \frac{K+1}{2}\right) + \mathbb{P}\left(X_K \geq \frac{K+1}{2}+1\right) + q^2\mathbb{P}\left(X_K \geq \frac{K+1}{2}-1\right)$$

and we obtain the desired inequality

$$\mathbb{P}\left(X_{K+2} \geq \frac{(K+2)+1}{2}\right) > \mathbb{P}\left(X_K \geq \frac{K+1}{2}\right).$$

To show the second part of the property, note that by the Weak Law of Large Numbers, for any $\epsilon > 0$

$$\lim_{K \to \infty} \mathbb{P}\left(|X_K/K - q| < \epsilon\right) = 1,$$

and because $q - X_K/K \leq |X_K/K - q|$,

$$1 = \lim_{K \to \infty} \mathbb{P}\left(|X_K/K - q| < \epsilon\right) \leq \lim_{K \to \infty} \mathbb{P}\left(q - X_K/K < \epsilon\right).$$

Since $\mathbb{P}\left(q - X_K/K < \epsilon\right) \leq 1$ it follows that

$$\lim_{K \to \infty} \mathbb{P}\left(q - \epsilon < X_K/K\right) = 1. \tag{14}$$

Because $q > 1/2$, there exists some $\epsilon > 0$ such that $q - \epsilon = \frac{1}{2}$ and that satisfies Equation (14):

$$1 = \lim_{K \to \infty} \mathbb{P}\left(\frac{1}{2} < \frac{X_K}{K}\right) = \lim_{K \to \infty} \mathbb{P}\left(X_K > \frac{K}{2}\right) = \lim_{K \to \infty} \mathbb{P}\left(X_K \geq \frac{K+1}{2}\right).$$

$\square$

**Lemma D.3.** *For any $n$ and non-negative sequences $\{a_i\}_{i=0}^n$ and $\{b_i\}_{i=0}^n$,*

$$\min_i \frac{a_i}{b_i} \leq \frac{\sum_{i=0}^n a_i}{\sum_{i=0}^n b_i}.$$

*Proof.* Let $r_{min} \triangleq \min_i \frac{a_i}{b_i}$. Then for any $i$, $\frac{a_i}{b_i} \geq r_{min}$, or equivalently

$$a_i \geq r_{min} b_i.$$

Hence,

$$\frac{\sum_{i=0}^n a_i}{\sum_{i=0}^n b_i} \geq \frac{\sum_{i=0}^n r_{min} b_i}{\sum_{i=0}^n b_i} = r_{min}.$$

$\square$

**Lemma D.4.** *For any $K \geq 3$,*

$$\frac{\sum_{y=0}^{\frac{K+1}{2}-2}\binom{K}{y}q^y(1-q)^{K-y} + \binom{K}{\frac{K+1}{2}-1}q^{\frac{K+1}{2}-1}(1-q)^{\frac{K+1}{2}}}{\sum_{y=0}^{\frac{K+1}{2}-2}\binom{K}{y}(1-q)^y q^{K-y} + \binom{K}{\frac{K+1}{2}-1}(1-q)^{\frac{K+1}{2}-1}q^{\frac{K+1}{2}}} > \frac{\sum_{y=0}^{\frac{K+1}{2}-2}\binom{K}{y}q^y(1-q)^{K-y}}{\sum_{y=0}^{\frac{K+1}{2}-2}\binom{K}{y}(1-q)^y q^{K-y}}.$$

*Proof.* It is equivalent to show that

$$\binom{K}{\frac{K+1}{2}-1}q^{\frac{K+1}{2}-1}(1-q)^{\frac{K+1}{2}}\sum_{y=0}^{\frac{K+1}{2}-2}\binom{K}{y}(1-q)^y q^{K-y}$$

$$> \binom{K}{\frac{K+1}{2}-1}q^{\frac{K+1}{2}}(1-q)^{\frac{K+1}{2}-1}\sum_{y=0}^{\frac{K+1}{2}-2}\binom{K}{y}q^y(1-q)^{K-y}$$

which reduces to

$$(1-q)\sum_{y=0}^{\frac{K+1}{2}-2}\binom{K}{y}(1-q)^y q^{K-y} > q\sum_{y=0}^{\frac{K+1}{2}-2}\binom{K}{y}q^y(1-q)^{K-y}$$

and equivalently

$$\frac{\sum_{y=0}^{\frac{K+1}{2}-2}\binom{K}{y}(1-q)^y q^{K-y}}{\sum_{y=0}^{\frac{K+1}{2}-2}\binom{K}{y}q^y(1-q)^{K-y}} > \frac{q}{1-q}.$$

The last inequality holds because by Lemma D.3,

$$\frac{\sum_{y=0}^{\frac{K+1}{2}-2}\binom{K}{y}(1-q)^y q^{K-y}}{\sum_{y=0}^{\frac{K+1}{2}-2}\binom{K}{y}q^y(1-q)^{K-y}} \geq \min_y \left(\frac{q}{1-q}\right)^{K-2y}$$

$$= \left(\frac{q}{1-q}\right)^{K-2\left(\frac{K+1}{2}-2\right)}$$

$$= \left(\frac{q}{1-q}\right)^3$$

$$> \frac{q}{1-q}.$$

$\square$