# OpenReview forum: "Counterbalancing Learning and Strategic Incentives in Allocation Markets"
_NeurIPS.cc/2021/Conference — NeurIPS 2021 Poster_

### Official Review · Reviewer_DTQS · 2021-07-16

**Rating:** 7
**Confidence:** 4

**Summary:**

The authors study a setting in which a central planner allocates a single indivisible good to agents showing up sequentially. The true quality of the item (good or bad) is unknown to both agents and the mechanism designer. The agents receive a signal about the quality of the item, and the planner aims to leverage those signals to learn the quality of the item and how to allocate it efficiently.

**Limitations And Societal Impact:**

The authors acknowledge the limitations of the paper. The current paper is one that could start an important line of work that will lead to relevant social and healthcare impact.

**Main Review:**

Strengths:
- The problem is very well and clearly motivated. From a practical point of view, the running example that the authors use is that of organ allocation. There, it is extremely important to ensure that organs are not discarded due to inefficiencies in online decision-making.
- The aspect studied by the authors is interesting. The point of view taken here is that agents learn from not only their private signals but also from past observations whether the organ is likely to be good or not, and base their willingness to get the item on this knowledge.
- The naïve sequential mechanism then does not work, because early agents believing the organ is bad lead all subsequent agents to believe the same, leading to the organ being discarded. I like the characterization of Lemma 1 that i) shows exactly when this can happen and ii) shows that this reduces to the beliefs of only the first 2 agents.
- The characterization of Theorem 1 is tight. If the information received by the agents is precise enough, then the authors develop a voting mechanism that is incentive-compatible and improves outcomes (correctness) compared to the naïve sequential mechanism. When not the case, they show that there is no truthful mechanism that will improve on the sequential one.
- The mechanism is relatively simple in principle and the idea behind it is natural (divide the population in increasing batch sizes and have them vote to decide on the allocation). It should be not too hard to implement in practice (up to getting the problem parameters right)
- The experiments complement the theoretical results well, showing that even just 2 batches already improve results significantly compared to the sequential algorithm.

Weaknesses:
- the model is still simple and first-cut, as acknowledged by the authors. The agents are fully symmetric (before they receive their signals), and all have the same utility/need for an organ.
- the randomness in the allocation may be undesirable, in the sense that an agent that is first of the remaining list might be skipped.
- What is the justification for agents having a private signal on the quality of the organ, versus there being a public signal? Why would individuals get different information about the organ in practice? It would be nice to talk about this a bit more.


**Time Spent Reviewing:**

3-4

---

### Official Review · Reviewer_MQKx · 2021-07-16

**Rating:** 6
**Confidence:** 4

**Summary:**

The authors study the problem of allocating a single indivisible item to a group of agents, without money. Motivated by kidney exchange, the item can only have two qualities: either good or bad. Every agent (as well as the mechanism designer) agree that the prior probability of the item good is $\mu = Pr[\text{item is good}]$. However, each agent i has their own private signal s_i about the item's quality. It is assumed that these signals are iid and depend on the true quality of the item: $q = Pr[s_i \text{ is good } |\text{ item is good}] = Pr[s_i \text{ is bad }|\text{ item is bad}]$, where this q is the accuracy of their signal. The utility of each agent is +1 upon receiving a good item and -1 for a bad one. The goal is to develop truthful mechanisms that maximize the probability that the item is correctly allocated: i.e., given to an agent if it is good or discarded if it is bad.

The authors study sequential allocation mechanisms, where the decisions of previous agents might reveal information about the item and affect following choices. This is empirically observed in Kidney Exchange, where if a kidney is rejected early, it is very likely that potential recipients might keep rejecting in, further decreasing the probability that someone will accept it.

The first result of the authors is to study the sequential offering mechanism. Here, the agents are ordered and the item is offered to each one, stopping and allocating the item the first time an agent accepts it. They show (roughly) that the item is allocated if $\mu > q$ or if the first two agents receive a good signal and it's never allocated for $\mu < 1-q$. Also, notice that his mechanism is not truthful. The reason the item might not get allocated is because the agents are actively updating their prior $\mu$.

The second (and main result) is the development of a *batched* version of the previous mechanism. In the simplest version, a batch of K agents is selected and asked to report their signals. If the majority considers the item good, then it is allocated at random to one agent reporting a good signal. Depending on $\mu$ and $q$, the authors show that there exists a range of batch sizes such that the mechanism is truthful. The second step is to greedily combine batches together: starting for a truthful batch size for $\mu$, if the item is not allocated then the next batch size would be a function of $\mu'$, the updated prior. Even though the game is sequential, the agents only participate once and only care about their current prior and the size of the batch they participate in.

Putting the two results together, the authors show that for $\mu > q$ there is no truthful mechanism and every mechanism is at most as good as the sequential one. This makes sense, as in this case the prior is more informative than the private signal. For $\mu < q$ however, their batched mechanism is incentive compatible and improves the guarantee. Finally, the authors have experiments on synthetic date showing the benefit of using 1 or 2 batches, compared to the sequential mechanism,



**Limitations And Societal Impact:**

The authors adequately explain the limitations and societal impact of their work. Kidney exchange is a very delicate matter and they acknowledge that although their paper is making progress towards the right path, randomization and batching users together might be perceived us unfair and be impractical: for example, imagine one recipient missing out because the rest of his batch voted against.

**Main Review:**

The paper's model is very clean and in my opinion original (although I am not an expert in social learning). The presentation is excellent, easy to follow and the authors clearly outline the limitations and modelling assumptions of their work (such as the reason every agent has the same $\mu$ and $q$).


The quality of the exposition and the mathematical rigor is high: although there isn't anything too technical going on, there is clever use of probability and Bayes' theorem to keep the analysis interesting and intuitive. My only recommendation would be to add an analysis for the case where $\mu$ and $q$ are not precisely known, but only within a confidence interval. I think most results would follow, but it would interesting to see if the new algorithm is robust to such noise. The sequential allocation mechanism performs worse in practice but does not require this information: would the batched mechanism still perform better with a slightly perturbed value of $\mu$ and $q$?

The significance of the paper is somewhat subjective: the paper provides a fresh model, with the right abstractions, that explains why there is inefficiency in kidney exchange and how it could be improved. I personally enjoyed it, but I could see why one might consider the technical contribution (and learning aspect) a bit weak.

**Time Spent Reviewing:**

4

---

### Official Review · Reviewer_ct3Q · 2021-07-17

**Rating:** 7
**Confidence:** 4

**Summary:**

This paper studies how to allocate an object with unknown quality (only prior is known) to strategic agents. The strategic agents have their own private signals (drawn i.i.d. from another common prior) about the quality of the object and they would update their beliefs after observing other agents' decisions.

The authors point out that sequentially offering the object would lead to a cascading effect; in particular, only the first two agents decide the outcome of allocation. In contrast, offering the object to all the agents to elicit their private beliefs all together would lead to an efficient allocation, if without taking strategic behavior into account. The authors propose a novel mechanism, which offers the object to agents batch-by-batch, leading to an improvement in the allocation performance. Their results are supported and validated using simulations.

**Limitations And Societal Impact:**

The main limitation of this paper is on the i.i.d. assumption for agents. The i.i.d. assumption, to some extent, assume that each agent is identical, which is probably not true in practice. For example, an agent may prefer an object more than another agent due to the level of compatibility. It would be very interesting to extend the results to the case in which the agents' priors are not identical.

**Main Review:**

This paper is well-written, clear, and easy to follow. The problem studied is nicely motivated by important real world applications. The collection of theoretical results is novel and interesting, which also gives interesting and useful insights to understand the problem. This work also opens many interesting follow-up questions to consider. The reviewer did not check all the proofs but they all look plausible in hindsight.

Comments:
1. It is surprising to see that two agents would determine the outcome in sequential offering mechanisms. The reviewer wonders perhaps this result is mainly due to the i.i.d. assumption. What would happen if the agents are not identical?

2. The reviewer really likes Lemma 2. It is very interesting to see that in order to achieve IC, the batch size cannot be too large or too small.

3. Given that the quality of the object is a binary, possibly the only relevant voting rule is majority. It would be interesting to see how to extend the results to an environment where the quality of the object is no longer a binary; and understand which voting rule could give the best outcome.

4. There is no theoretical guarantee on how the greedy voting mechanism performs. It would be great to figure out whether the greedy voting mechanism is indeed optimal; if no, then understand what approximation does the mechanism give and also find out the optimal mechanism.

5. Figure 3 is confusing. Is V_GREEDY and V_OPT the same? Could we try V_GREEDY^K with K > 2? Is it better than V_GREEDY^2?

Minors:
*. After reading Lemma 2, the reviewer is curious to see a table or a plot of upper and lower bounds. This is shown in Figure 2 but it might be better to provide a reference to the figure here so that the readers do not need to wonder about the bounds.

**Time Spent Reviewing:**

2

---

### Official Review · Reviewer_2Yd7 · 2021-07-20

**Rating:** 6
**Confidence:** 3

**Summary:**

This paper presents an idea to improve the correctness of an allocation when the quality of the good must be inferred from agents who learn from each other. The authors set up a problem of allocating one object to a group of agents. The object is either of good or bad quality and each agent has a noisy IID signal of the quality. Together, they have a pretty good estimate of the quality. But each agent may have reasons to withhold their signal in order to avoid getting a bad object or attempt to acquire a good object. With incentives and learning together, the allocation can fail badly using a standard approach of offering the item to each agent sequentially to either accept or reject. The authors prove that, in their simplified model, only the first two agents play a role. If both of the first two agents reject the object, then every agent will reject the object, since they learn about quality from each other. Observing two rejections is enough to update every agents' belief to below threshold. The authors propose instead a clever algorithm. The planner offers the object to a batch of agents at a time. The batch size is calculated to maximize correctness subject to incentive compatibility. If every agent in the first batch rejects the object, the object is offered to a new, larger, batch of agents. The authors prove that, if the signals are more informative than the prior, then their mechanism is incentive compatible and is significantly more correct (has higher social welfare) than the sequential offer mechanism. The authors also prove that if the prior is more informative than the signals, then no mechanism can do better than the sequential offer mechanism.

**Limitations And Societal Impact:**

Yes, although the authors may have overstated the realism of their approach for reducing discards in organ donation.

**Main Review:**

The paper provides a clever algorithm to solve a stylized version of an interesting and realistic problem. The authors do a fantastic job of providing explanations and intuition behind their proofs and math. The algorithm is a satisfying solution. The paper is a good illustration of how incentives and learning can collide to interfere with one another, and provides a natural solution that will probably generalize to less stylized cases.

I generally like the paper, but I have several quibbles. First, using organ donation as the primary motivator, highlighted in the first sentence of the abstract, seems like a stretch. The stylized model does not really fit the organ donation example. In organ donation, organs are not just "good" and "bad" for everyone. The quality of an organ for an agent is highly personalized, according to the level of match with blood type as the highest-order bit and with many other factors. If an organ is not good for one patient, it should mean little for the next patient. Also, patients themselves don't have the best information about the quality of an organ: doctors and hospitals do. Presumably, if the organ is "bad", the hospital system will not offer it to anyone. Finally, the organ application seems like it is thrown in just to draw interest: the authors make no attempt to use real data from organ exchanges in the paper and don't really discuss how their stylized model diverges from the application complexities. The paper is a theory paper with simulations: I encourage the authors to be more clear about the strength of fit to applications.

Real estate might be a better application. When a house is on the market for a long time without being sold, buyers may think, "something must be wrong with the house". Agents do have information about the house. It's still not perfect, since real estate values are also subjective, and since the real estate market involves prices. In general, I suggest the authors not highlight an application in detail unless some real data is used or care is taken to discuss the fit of theory to reality. Potential applications can be mentioned in the intro, but I wouldn't highlight one of them in the first sentence of the abstract.

Second, I'd like more discussion of how important key assumptions are to the analysis. In particular, the fact that the true positive rate equals the true negative rate (line 97) seems like a strong and unrealistic assumption.

Third, the simulations don't add a whole lot, in part because the theory is strong. Using some kind of real data would make the paper much stronger than running simulations.

Why isn't V_GREEDY one of the simulated algorithms? Why are only the 1- and 2-batch algorithms simulated? This seems strange. The main algorithm in the paper is V_GREEDY. I would expect this to be simulated if nothing else.

When defining incentive compatibility (line 127), I suggest the authors mention that agents will update u as they learn from other agents.

In Figure 3, how does V_ALL achieve perfect correctness? Presumably, even V_ALL will allocate incorrectly with some small probability, if, for example, many agents receive an incorrect signal by random chance.

Some minor suggestions:

thousands of future lives.
thousands of lives.

Line 45: "the list" -->
"the waiting list", or "the list of agents"

=====
After author response:

I read the authors' response and the other reviews. The authors addressed my concerns well. I don't think it's a slam-dunk paper, but I do think it's worthy of being accepted, especially given the unusual consensus (on ratings) among the reviewers. It was the best submission among those that I reviewed.

**Time Spent Reviewing:**

4

---

### Author Response · Authors · 2021-08-10
**Author response**

We thank the reviewers for very insightful comments.

First, we want to address some of the common feedback regarding organ allocation application. We acknowledge that our model is highly stylized with several abstractions. We plan to further emphasize this fact and clarify the key differences with the organ allocation application.
* *Common utility function*:  In practice, organ allocation waitling list have patients of varying medical characteristics (e.g., age, blood type), which may give rise to heterogeneous utility from the same organ. Our model currently assumes a common utility function, which can be thought of restricting attention to patients within the same medical characteristics group. Model extension to account for heterogeneous utility function would be an interesting and useful future direction. Results from the herding literature show that heterogeneous utility rarely affects the overall release of information, although the convergence process can be slower (Smith and Sorensen (2000)). We believe that our qualitative findings will still hold, but further analysis is required.

* *Private information*: In our motivating example, we presumed that patients obtain private information about an organ from their own doctors (lines 36-38). In fact in most cases, doctors directly make the accept/reject decisions on behalf of their patients. Several recent medical studies have shown that there is a significant variability across healthcare providers in organ quality assessment (e.g., different interpretation of biopsy) (see King et al. (2020), Husain et al. (2020) and Emmons et al. (2021)). We will make this assumption much clearer in the next version.

* *Signal precision* $q$: Based on the social learning literature (e.g., Smith and Sorensen (2000)), we believe that our results will be robust even if the signal structure is not symmetric (each agent has different $q$) and/or more elaborate (different true positive and true negative rates) as long as signals are binary and bounded (i.e., signal is never perfect). We believe that a rejection cascade will still take place in the sequential offering mechanism and that there will be a correctness-improving voting mechanism. Thank you for pointing out this direction for follow-up work.

Now we address other important concerns raised by each reviewer.

**Reviewer 2Yd7:**
- Using empirical data instead of simulation is a great suggestion --  we have started exploring a follow-up project in this direction. We also agree with the reviewer's point that this is a theory paper. As suggested, we will downplay the applications in the next version of our abstract.

- Regarding simulations: We initially included only $V_{GREEDY}^1$ and $V_{GREEDY}^2$ to showcase that greedy voting mechanisms can achieve improvement with only 1 or 2 batches, which could also be more realistic and serve as lower bounds on the improvement of $V_{GREEDY}$ (by Proposition 3 (iii), line 295). However, we agree with the reviewer's comment that simulating the main algorithm, $V_{GREEDY},$ would be more useful. Also in Figure 3, as pointed out in the comment, $V_{ALL}$ achieves near perfect correctness but not exactly $1.$ We will make this observation clearer in the caption.

**Reviewer ct3Q:**
- Thank you for recognizing the novelty of our paper. We also believe that our paper may open many interesting follow-up works. Regarding the extension to non-binary signals (and equivalently, non-binary state space): this is an interesting direction but can have some challenges (e.g., how to aggregate preferences, rank order among multiple states).

- In Figure 3, there was a typo: $V_{OPT}^1$ is $V_{GREEDY}^1$ and $V_{OPT}^2$ is $V_{GREEDY}^2.$ We will fix it in the next version.

**Reviewer MQKx:**
- Regarding perturbed $\mu$ and $q$: this is a very interesting direction. As agents make decisions based on expected utility, we believe that the addition of such noise will not alter the flavor of the results. The improvement achieved by voting mechanisms could indeed be smaller.

**Reviewer DTQS:**
- Regarding the common utility and justification for private signals, please see our response to common feedback (at the top of our response).

- Regarding the voting mechanism's potential unfairness caused by randomness, we agree that it is a limitation -- as noted, our solution is not a Pareto improvement for every agent (lines 379-380). To follow up, we plan to quantify this unfairness by characterizing how much of the waiting list's predetermined priority a voting mechanism violates in expectation.

---

> ### Comment · Reviewer_DTQS · 2021-08-22
> **Thanks a lot for the response!**
>
> I wanted to thank the authors for responding to the reviews, and for acknowledging some of the abstractions of the work! My overall opinion, which is the paper studies an important problem, makes a nice and non-trivial contribution, and should be accepted, has only been reinforced. I also appreciate that the authors have put some serious thoughts into how to build up from the current paper to a possible less abstract and more realistic model.

---

### Decision · Program_Chairs · 2021-09-27

**Decision:**

Accept (Poster)

**Comment:**

This paper is an easy and uncontroversial accept.